# LANG-PINN: FROM LANGUAGE TO PHYSICS-INFORMED NEURAL NETWORKS VIA A MULTI-AGENT FRAMEWORK

## ABSTRACT

Physics-informed neural networks (PINNs) provide a powerful approach for solving partial differential equations (PDEs), but constructing a usable PINN remains labor-intensive and error-prone. Scientists must interpret problems as PDE formulations, design architectures and loss functions, and implement stable training pipelines. Existing large language model (LLM) approaches address isolated steps such as code generation or architecture suggestion, but typically assume a formal PDE is already specified and therefore lack an end-to-end perspective. We present **Lang-PINN**, an LLM-driven multi-agent system that builds trainable PINNs directly from natural language task descriptions. **Lang-PINN** coordinates four complementary agents: a *PDE Agent* that parses task descriptions into symbolic PDEs, a *PINN Agent* that selects architectures, a *Code Agent* that generates modular implementations, and a *Feedback Agent* that executes and diagnoses errors for iterative refinement. This design transforms informal task statements into executable and verifiable PINN code. Experiments show that **Lang-PINN** achieves substantially lower errors and greater robustness than competitive baselines: mean squared error (MSE) is reduced by up to 3–5 orders of magnitude, end-to-end execution success improves by more than 50%, and reduces time overhead by up to 74%.

## 1 INTRODUCTION

Partial differential equations (PDEs) are central to scientific computing, underpinning applications in physics, engineering, and materials science. Physics-informed neural networks (PINNs) (26) have emerged as a flexible framework that embeds governing equations into trainable neural models, offering a unified approach for forward, inverse, and data-scarce problems (12; 22). Despite their promise, training PINNs remains highly challenging: they suffer from gradient pathologies (36), ill-conditioning from the neural tangent kernel perspective (37), failure modes in complex regimes (13), and sensitivity to activation functions, sampling, and decomposition strategies (10; 46; 42; 30; 9). Although libraries and benchmarks, such as DeepXDE (22), PINNacle (7), and PDEBench (34), have been developed to to solve these problems , deploying a trainable PINN still requires expert-level manual effort in PDE specification, architecture design, and optimization tuning.

Efforts to reduce this burden remain fragmented. Traditional automation focuses on hyperparameter search (32; 16; 6; 8) or architecture variants (30; 40; 39), but these approaches assume that the governing PDE has already been written down in an explicit and computationally usable form. Recent progress in large language models (LLM) enables natural-language interfaces to computational tools, including code generation (27; 17; 21) and multi-step reasoning (45; 29; 23; 38; 41). Domain-specific prototypes such as CodePDE (18) and PINNsAgent (43) show that LLM-driven PDE solvers are feasible, but they still require manually defined PDE schemas or provide limited verification and iterative refinement. As a result, current automation begins only after the PDE has been fully specified and provides no assistance for constructing the equation itself. This limitation is substantial because designing or revising a PDE is often the most technically demanding part of developing a PINN. It requires precise reasoning about operators, coefficients, and boundary or initial conditions, and even small changes in the scientific setting can lead to meaningful adjustments of the equation. In contrast, describing a new configuration or an updated setting in natural language

is straightforward for researchers. For example, expressing that "the heat source is moved from the center of the domain to the boundary" is simple in text, yet it alters the source term, the boundary conditions, and the spatial dependence of the governing equation. The contrast between the ease of expressing such scientific changes in language and the difficulty of updating the corresponding PDE reveals a clear gap: existing systems lack a mechanism that links natural-language descriptions to the fully specified equations required for PINN training. This gap motivates the development of automated text-to-PDE construction.

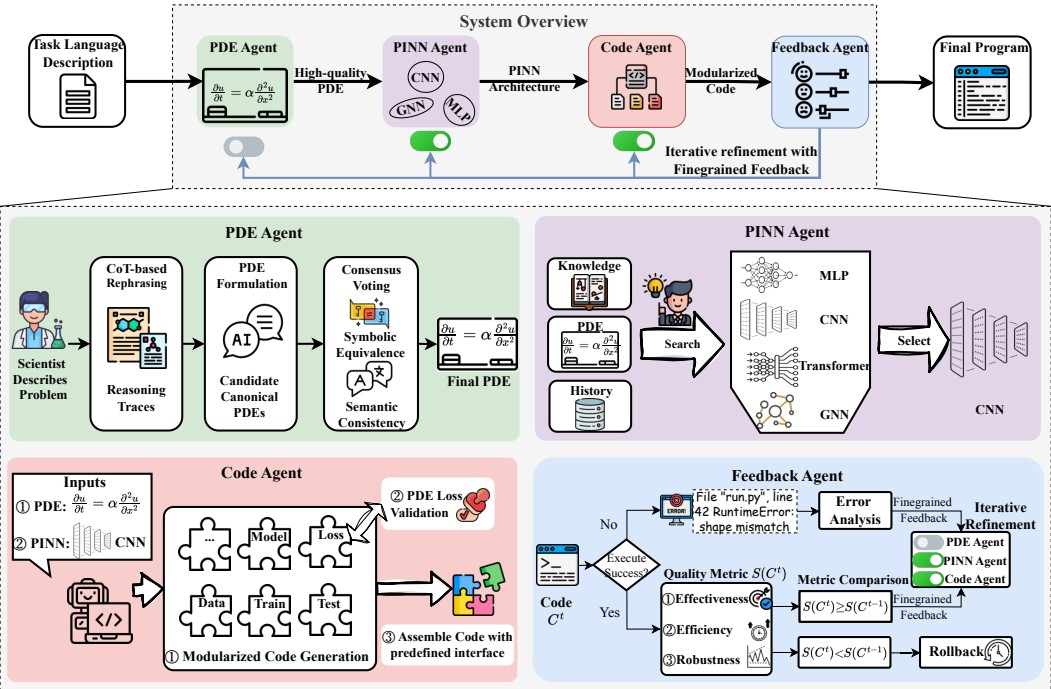

Figure 1: System overview of **Lang-PINN**. The framework decomposes end-to-end PINN design into four agents: *PDE Agent* (canonical PDE formulation), *PINN Agent* (training-free architecture selection), *Code Agent* (modularized code generation), and *Feedback Agent* (runtime error analysis and multi-dimensional evaluation). Iterative refinement with feedback forms a closed loop, yielding reliable and executable PINN programs from natural language descriptions.

To address this gap, we propose a multi-agent framework, namely **Lang-PINN**, that decomposes the workflow into four cooperating roles, as shown in Fig. 1. The *PDE Agent* formulates natural language into operators, coefficients, and boundary/initial conditions. The *PINN Agent* aligns PDE characteristics—periodicity, geometric complexity, and multiscale or chaotic dynamics—with inductive biases via a requirement vector and utility score. The *Code Agent* generates modular, contract-preserving training code, while the *Feedback Agent* executes the code, monitors residuals and convergence, and iteratively guides corrections. This structured, verifiable pipeline ensures that scientific consistency, executability, and trainability are treated as first-class design goals.

Our contributions are as follows:

- We propose the first framework that starts directly from natural language task descriptions and automatically produces complete PINN solutions, including PDE formulations, architecture selection, code generation, and feedback-driven refinement, thereby lowering the entry barrier for domain scientists.

- We construct a benchmark dataset that pairs four-level difficulty task descriptions with ground-truth PDEs, enabling systematic evaluation of semantic-to-symbol grounding and supporting verifiable, reproducible PINN design.

- We demonstrate that our multi-agent framework achieves substantial improvements across diverse PDEs, reducing mean squared error by up to *3–5 orders of magnitude*, increasing

code executability success rates by more than *50%*, and reducing time overhead up to *74%* compared to strong agent-based baselines.

## 2   RELATED WORK

**Physics-Informed Neural Networks.**   Physics-Informed Neural Networks (PINNs) (26) integrate governing equations into neural training by penalizing PDE residuals and boundary violations. Numerous variants improve convergence and accuracy through adaptive activations (10), gradient-enhanced residuals (46), adaptive sampling (22; 42), or domain decomposition (30; 9). Yet, these approaches still require experts to manually specify PDE formulations, architectures, and loss terms. Our work instead seeks to automate these design choices from task descriptions.

**LLM Agents and Reasoning Strategies.**   Large language and code models have enabled text-to-code generation (27; 17) and agentic software engineering (11; 44). In scientific domains, Code-PDE (18) demonstrates that inference-time reasoning and self-debugging can produce PDE solvers directly from text. Complementary prompting strategies such as SCoT (15) and Self-Debug (3) improve logical consistency and error correction through structured reasoning or iterative reflection. However, these remain single-agent methods without physics-grounded validation, limiting their applicability to scientific surrogates. Our framework extends this direction by coupling reasoning and feedback across multiple specialized agents tailored to PINNs.

**Automated PINN Design.**   Classical Automated Machine Learning (AutoML) methods (8), including Bayesian optimization (32), Hyperband (16), and BOHB (6), aim to reduce manual effort in tuning architectures and hyperparameters. Applied to physics-informed settings, however, they struggle with residual imbalance, unit inconsistency, and multi-scale stiffness, often requiring expert intervention. Recent PINN-oriented searches (36; 42) mitigate some challenges but still assume human-specified PDEs and loss structures. In contrast, our approach introduces a dedicated multi-agent system for PINN automation, integrating PDE translation, architecture design, and feedback-driven refinement to minimize manual design effort and achieve end-to-end trainability.

## 3   AN INVESTIGATION ON MODULES OF PINNS

Despite recent progress concentrating on PDE parsing and PINN architecture search, the properties of the modules in PINNs remain under-explored. Since the ultimate goal of this paper is to build an end-to-end, automated PINN pipeline, it is crucial to obtain a comprehensive understanding of these modules in PINNs. To this end, in this section, we conduct a series of empirical analyses on three pivotal modules in PINN pipelines, including PDE parsing, architecture curation, and code generation, and demonstrate that existing PINN pipelines suffer from three bottlenecks in practice: problem formulation linguistic variability, model performance variability, and code generation complexity.

### 3.1   LINGUISTIC VARIABILITY OF TASK FORMULATION FROM TEXTUAL DESCRIPTION TO PDES

Typically, a PINN pipeline begins with translating natural-language descriptions into formal PDEs. In the generated PDEs, the loss terms are defined, the solution space is constrained, and all downstream stages are conditioned. As the foundation of the entire pipeline, any error in this step invalidates the pipeline. Thus, it is essential to formulate the PDEs in a reliable way.

In this section, we propose to determine the significance of the reliable PDE formulation. Specifically, we propose a tiny augmented dataset, *Task2PDE*, where eight examples are randomly sampled from PINNacle benchmark (7) and re-expressed with four levels of linguistic variability. Details about re-expression are available in Appendix 4. In this way, each sample is paired with 50 descriptions for each level of variability, yielding 1,600 description-PDE pairs. We adopt four popular open-source LLMs (Llama2 (35), Vicuna (4), DeepSeek-V3 (5), Qwen (1)) and evaluate them with *symbolic equivalence* over the Task2PDE dataset. Results in Fig. 2 show that symbolic accuracy declines steadily as the linguistic variability of the descriptions increases, indicating that even small shifts in wording can substantially alter the PDE inferred by an LLM and undermine the reliability of the formulation.

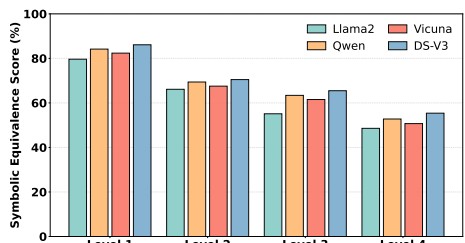

Although symbolic equivalence provides a mathematically precise way to verify PDE forms, it is overly brittle when applied directly to natural-language derived expressions: mathematically identical terms (e.g., $u_{xx}$ vs. $\partial^2 u/\partial x^2$) are flagged as mismatches, and benign coefficient variations are misclassified. These limitations do not undermine the value of symbolic checking itself, but rather indicate that symbolic matching cannot be the sole validation mechanism when inputs are noisy or stylistically diverse. This motivates the *PDE Agent* (Sec. 4.2), which augments symbolic checks with semantic evaluation and consensus voting, enabling robust PDE formulation while still benefiting from symbolic verification as a final correctness safeguard.

Figure 2: Impact of linguistic variability on PDE translation. Accuracy is reported across four levels of description difficulty using symbolic equivalence.

### 3.2 VARIABILITY OF ARCHITECTURE PERFORMANCE ACROSS PDEs

Once the PDE is specified, selecting a suitable PINN architecture is crucial. The inductive bias of the network, such as its preference for local patterns, long-range dependencies, or structural constraints, directly affects stability and accuracy. A poor match can lead to slow convergence or large residual errors. To demonstrate this effect, we benchmark four representative architectures (MLP, CNN, GNN, and Transformer) on PDEs including Shallow Water, Convection, Poisson, and Heat. As shown in Fig. 3, performance varies markedly across PDEs. CNNs and Transformers excel on Convection and Heat, while MLPs and GNNs achieve the lowest error on Poisson. For Shallow Water, differences are minor. These results

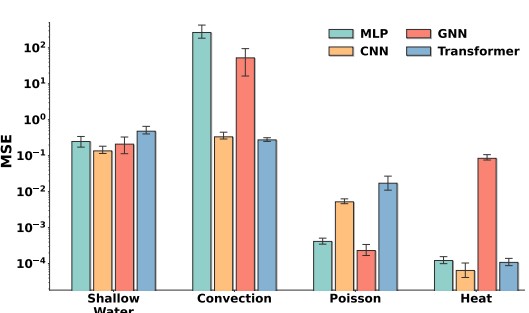

Figure 3: Comparative MSE of different PINN architectures on representative PDEs. Results are shown in log scale for clarity.

show that no single architecture is universally effective, motivating approaches that adapt PINN designs to the operators and structures of different PDEs.

### 3.3 COMPLEXITY OF CODE GENERATION IN END-TO-END WORKFLOWS

After the PDE and PINN architecture are specified, the next step is to generate executable code, including model definitions, physics-informed losses, data pipelines, and training routines. This process is complex because multiple components must not only be correct in isolation but also interact reliably, making executability a central challenge.

To study code generation paradigms, we compare *monolithic generation*, where an LLM produces the entire pipeline in a single pass, with *modular generation*, where code is synthesized by components. As shown in Fig. 4, modular generation consistently achieves more than twice the success rate of mono-

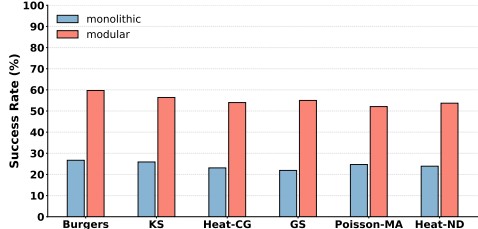

Figure 4: Comparative Success Rate(%) of different code generation paradigm (monolithic vs. modular) on six PDEs.

lithic generation across six representative PDEs (Burgers, KS, Heat-CG, GS, Poisson-MA, Heat-ND). The modular design localizes errors, preserves correct components, and avoids regenerating the full script, thereby substantially improving executability. These results motivate the design of the **Code Agent**, which adopts the modular paradigm. We note that this experiment isolates the effect of modularization alone; when combined with the **Feedback Agent** in our full framework, success rates improve even further, as shown in later sections.

## 4 METHOD

### 4.1 SYSTEM OVERVIEW

According to the analyses in the previous section, we observe that end-to-end PINN automation breaks down due to cascading dependencies: minor linguistic variations in PDE formulation propagate into architecture mismatches, and monolithic code generation further amplifies upstream errors. Thus, in this section, we propose to construct an end-to-end automated PINN framework to enhance verification and reliability.

Fig. 1 presents **Lang-PINN**, our multi-agent framework that converts natural-language task descriptions into executable PINN training code. It consists of four agents with distinct roles: the *PDE Agent* formalizes task descriptions into governing equations, the *PINN Agent* selects suitable architectures, the *Code Agent* generates modular implementations, and the *Feedback Agent* executes and evaluates outputs. These agents interact in a sequential workflow, with the *Feedback Agent* providing iterative diagnostics that refine earlier stages, particularly code generation. This modular and feedback-driven design reduces error propagation and ensures reliable, scientifically valid PINN implementations.

### 4.2 PDE AGENT

To alleviate the sensitivity to linguistic variability identified in Sec. 3.1, the *PDE Agent* uses a label-free reasoning–selection pipeline. Given a task description $d$, the agent samples $K$ chain-of-thought (CoT) trajectories, cleans each trajectory into a normalized description $\hat{d}_k$, and formulates a canonical PDE candidate $E_k$. Invalid candidates are filtered by template validation (operator well-formedness, residual form, admissible boundary/initial terms). The remaining set $\mathcal{E} = \{E_1, \ldots, E_K\}$ is then resolved via consensus voting, and the agent selects the candidate that is most similar to the others under a joint symbolic–semantic criterion.

**Symbolic Equivalence.** To assess whether two candidate PDEs express the same operator structure, we compute a *symbolic equivalence score* based on their abstract syntax trees (ASTs). Each PDE $E$ is parsed into a canonical symbolic tree $\mathcal{T}(E)$ using Sympy, where nodes represent operators (e.g., $\partial_t$, $\partial_x^2$, nonlinear products) and leaves correspond to variables or constants.

Given two trees $\mathcal{T}(E_i)$ and $\mathcal{T}(E_j)$, we define their symbolic equivalence as a normalized tree-matching score,

$$\mathrm{sym}(E_i, E_j) = \frac{|\mathcal{M}(\mathcal{T}(E_i), \mathcal{T}(E_j))|}{\max(|\mathcal{T}(E_i)|, |\mathcal{T}(E_j)|)}, \tag{1}$$

where $\mathcal{M}(\mathcal{T}(E_i), \mathcal{T}(E_j))$ denotes the set of matched subtrees under operator-preserving alignment, and $|\mathcal{T}(\cdot)|$ counts the total nodes. This yields a score in $[0, 1]$, equal to 1 if two PDEs are symbolically equivalent (identical operator trees) and decreasing smoothly as structural discrepancies grow.

This formulation abstracts our Sympy-based implementation, where equivalence is resolved by recursively comparing operator nodes and their children up to commutativity and normalization rules. It aligns with symbolic regression principles (28; 14), while providing robustness to variations in coefficient presentation or term ordering.

**Semantic Consistency.** Symbolic matching alone cannot capture cases where mathematically equivalent PDEs are expressed in different notations or variable names. Following ideas from mathematical information retrieval (47), we therefore introduce a *semantic consistency* score. Each candidate PDE $E$ is paraphrased into a normalized summary $g(E)$ that encodes its domain, operator types, and forcing terms. The semantic consistency between two candidates $E_i$ and $E_j$ is then defined as

$$\mathrm{sem}(E_i, E_j) = \sigma\big(g(E_i), g(E_j)\big), \tag{2}$$

where $\sigma$ is a sentence-level similarity function such as embedding cosine similarity or LLM-based entailment scoring. This yields values in $[0, 1]$ and provides robustness to symbol renaming, coefficient scaling, or algebraic rearrangements that preserve meaning but alter surface form.

**Consensus Voting.** Finally, we combine symbolic and semantic similarities into a composite score $S(E_i, E_j) = \alpha \, \mathrm{sym}(E_i, E_j) + (1 - \alpha) \, \mathrm{sem}(E_i, E_j)$. Each candidate is then compared against the

others, and the one with the highest average similarity is selected as the final PDE. This simple consensus step ensures that the chosen equation is both structurally consistent and semantically faithful to the task description. We use a calibrated similarity threshold of 0.80; details are provided in Appendix 3.4.

## 4.3 PINN AGENT

Different PDEs exhibit distinct sensitivities to network architecture, and no single model is uniformly optimal (Sec. 3.2). The *PINN Agent* selects an appropriate architecture for a newly extracted PDE without any training-time search. Given a canonical PDE representation $E$, the agent follows a two-stage process: it first queries a *history cache* $\mathcal{H}$ to reuse the architecture and hyperparameters of a previously solved, highly similar PDE; if no such entry exists, it performs knowledge-guided matching using a *knowledge base* $\mathcal{K}$, which scores architectures by their compatibility with the characteristics of $E$. This design enables efficient reuse for recurring PDEs while providing principled generalization to unseen ones.

**History Reuse.** The history cache $\mathcal{H}$ is an automatically maintained collection of past tasks. Each entry records: 1) the natural-language task description, 2) the extracted PDE together with its feature vector $\phi(E)$, and 3) the architecture and hyperparameters that produced the best PINN solution.

**Knowledge-guided Matching.** In the absence of reusable history, the agent applies *knowledge-guided matching* to select architectures based on knowledge base $\mathcal{K}$. The key idea is to embed PDEs and architectures into a representation vector, where their alignment can be systematically evaluated. We first describe how PDEs are represented, then how architectures are encoded, and finally how the two are matched.

*1. PDE Feature Representation.* To represent the input side of the matching process, each PDE $E$ is encoded as a feature vector

$$\phi(E) = [f_1(E), f_2(E), \ldots, f_n(E)]^\top, \tag{3}$$

where $f_i(E)$ denotes a quantifiable physical property, including *periodicity*, *geometry complexity*, and *multi-scale demand*. Periodicity reflects whether domains or boundary conditions repeat, geometry complexity captures whether the domain is structured or irregular, and multi-scale demand indicates the extent of interacting scales or chaotic regimes. Formal definitions are given in Appendix 1. These dimensions are motivated by prior findings that Fourier or sinusoidal layers align with periodic problems (31; 19), graph-based models are effective for irregular geometries (24; 2), and attention or spectral operators handle multi-scale demand, e.g., dynamics (25).

*2. Architecture Capability Representation.* To make architectures comparable with PDE features, each architecture $\mathcal{A}$ is represented by a capability vector

$$\psi(\mathcal{A}) = [a_1(\mathcal{A}), a_2(\mathcal{A}), \ldots, a_n(\mathcal{A})]^\top, \tag{4}$$

where $a_i(\mathcal{A})$ measures its competence on property $i$ within PDE feature representation. Capability values are inferred through LLM reasoning and refined with historical experimental outcomes, ensuring adaptability across tasks. Formal definitions are given in Appendix 1.

*3. PDE–Architecture Matching* The compatibility between a PDE $E$ and an architecture $\mathcal{A}$ is measured using a weighted cosine similarity:

$$S(\mathcal{A}, E) = \frac{(\mathbf{W}\phi(E))^\top \psi(\mathcal{A})}{\|\mathbf{W}\phi(E)\|_2 \cdot \|\psi(\mathcal{A})\|_2}, \tag{5}$$

where $\mathbf{W} = \mathrm{diag}(w_\mathrm{per}, w_\mathrm{geo}, w_\mathrm{ms})$ assigns importance weights to each property. In practice, we prioritize multi-scale demand over geometry and periodicity, as mismatches on the former are most detrimental to convergence (19; 24; 2; 31). The final architecture is then selected as

$$\mathcal{A}^\star = \arg\max_{\mathcal{A}\in\Theta} S(\mathcal{A} \mid E). \tag{6}$$

**Architecture Template Instantiation.** After selecting $\mathcal{A}^\star$, the agent instantiates the model via predefined *architecture templates*. Each template specifies the essential architectural parameters,

such as the number of layers, hidden width, activation function, embedding dimension, and the training hyperparameters, including learning rate, batch size, and optimizer. These templates expose their parameters as fillable fields, which are populated using the configuration stored in $\mathcal{K}$ for $\mathcal{A}^\star$ (or default entries when unavailable). This templated instantiation avoids errors from free-form code generation and ensures consistent, reproducible construction of PINN models across tasks.

## 4.4 CODE AGENT

Directly prompting an LLM to generate the entire PINN pipeline in one pass often produces brittle code, where model definition, loss formulation, and training loops are tightly coupled. Errors become difficult to isolate, and fixing them typically requires regenerating the whole script. To avoid this, the *Code Agent* adopts a modular strategy with explicit verification mechanisms.

**Modularized code generation.** Instead of producing a monolithic script, the *Code Agent* decomposes the pipeline into independent modules: (i) model definition, (ii) PDE loss, (iii) data pre-processing, (iv) training loop, (v) validation, and (vi) main function. Each module is generated separately, allowing faults to be localized and corrected without regenerating unrelated components.

**Interface constraints.** Modules are connected through standardized input–output formats, ensuring compatibility and composability. This design makes it possible to update or replace one module without introducing inconsistencies elsewhere, thereby reducing correction cost and enabling fine-grained refinement.

**PDE loss verification.** For the PDE loss module, the generated code is parsed back into a symbolic PDE $\hat{E}$ and checked for equivalence with the PDE $E$ provided by the *PDE Agent*. Only loss modules that pass this symbolic check are retained, ensuring that the optimization objective faithfully encodes the governing equation.

## 4.5 FEEDBACK AGENT

The *Feedback Agent* closes the loop by leveraging runtime signals to refine earlier stages. Built on the modular code of the *Code Agent*, it translates execution diagnostics into localized suggestions, avoiding global regeneration and improving reliability.

**Error localization and correction.** When executing the generated code, two scenarios arise. If runtime errors occur, the *Feedback Agent* analyzes the error messages and attributes them to the most likely module (e.g., model structure, loss function, training loop). It then instructs the *Code Agent* to regenerate only the faulty component, avoiding unnecessary changes to other modules. If the issue originates upstream (e.g., in PDE specification or PINN architecture), the *Feedback Agent* can escalate its directive to the corresponding agent, ensuring that corrections are applied at the appropriate level.

**Multi-dimensional quality evaluation.** If execution succeeds, the *Feedback Agent* evaluates the code along three complementary dimensions: (i) *effectiveness*, measured by PDE residual error (e.g., MSE); (ii) *efficiency*, measured by convergence speed and resource cost (steps, FLOPs, parameters); and (iii) *robustness*, measured by loss smoothness and the absence of gradient pathologies. Each metric is normalized, and a weighted sum produces an overall quality score:

$$S(C) = \sum_{i=1}^{3} w_i \, \hat{m}_i(C), \tag{7}$$

where $C$ denotes the generated code, $\hat{m}_i$ the normalized value of the $i$-th metric, and $w_i$ its weight. Detailed definitions and quantification of these metrics are provided in Appendix 2.

**Iterative refinement.** The decision to accept or reject a new version is based on comparing the current score $S(C^{(t)})$ with the previous score $S(C^{(t-1)})$. If the new version improves, the agent proceeds; otherwise, it reverts and restarts optimization. By coupling modular generation with runtime feedback, the system ensures that diagnostic signals can be acted upon locally rather than globally, providing fine-grained corrections that improve reliability and efficiency over iterations.

## 5 EXPERIMENTS

### 5.1 EXPERIMETAL SETTINGS

**Benchmark Datasets**   We evaluate **Lang-PINN** on the PINNacle benchmark (7), which comprises 14 representative PDEs across 1D, 2D, 3D, and ND settings: *Burgers, Wave-C, KS, Burgers-C, Wave-CG, Heat-CG, NS-C, GS, Heat-MS, Heat-VC, Poisson-MA, Poisson-CG, Poisson-ND, Heat-ND*. This collection spans diverse dimensionalities, geometric complexities, and dynamical regimes, providing a rigorous testbed for automated PINN design. At the task-to-PDE stage, **Lang-PINN** operates from natural-language inputs: for each PDE we construct three distinct textual problem descriptions, which must be translated into canonical PDE formulations before downstream modeling. In contrast, baseline methods cannot perform this translation step and are therefore provided directly with the canonical PDE formulations from the benchmark. For fairness, all quantitative metrics are computed solely on the resulting PINN performance, independent of whether the PDE was inferred or given. Each task is evaluated over 10 independent runs, and within each run the agent is allowed up to three refinement iterations, ensuring both fairness across methods and robustness to stochasticity in generation.

**Baselins**   We include **PINNacle** (7) as a non-agent reference that fixes both PDEs and architectures and directly trains PINNs. All other baselines adopt LLM-based agent but still assume the PDE and architecture are given. **RandomAgent** and **Bayesian-Agent** explore architectures through random or Bayesian search with error-only feedback, while **SCoT** (15), **Self-Debug** (3), and **PINNsAgent** (43) rely on prompting to generate losses or partial code, again without full feedback or PDE formulation. As summarized in Table 1, none of these baselines support PDE formulation, code generation is at best partial, and feedback is limited to error detection, whereas **Lang-PINN** spans all dimensions in a coordinated multi-agent system. We adopt Deepseek-V3 (5) (top-p=0.9, temperature=0.2, max_tokens=2048) as the LLM backbone for all agent-based baselines and our Lang-PINN for a fair comparison.

Table 1: Comparison of methods across five functional dimensions: **PF** (PDE formulation), **AD** (architecture design), **CG** (code generation), and **FS** (feedback signal). For feedback signal, "Err+Metrics" augments runtime error with validation metrics.

| Method | PF | AD | CG | FS |
|---|---|---|---|---|
| PINNacle | ✗ | ✗ | ✗ | ✗ |
| RandomAgent | ✗ | ✓ | Partial | ✗ |
| BayesianAgent | ✗ | ✓ | Partial | ✗ |
| SCoT | ✗ | ✗ | Partial | ✗ |
| Self-Debug | ✗ | ✗ | Partial | Err-only |
| PINNsAgent | ✗ | ✓ | Full | Err+Metrics |
| **Lang-PINN** | ✓ | ✓ | Full | Err+Metrics |

**Metrics** The **success rate** measures robustness by reporting the proportion of runs in which the generated code executes end-to-end without runtime errors, independent of training accuracy. The **mean squared error (MSE)** quantifies numerical fidelity of the resulting PINN solution. The **iterations to a successful run** capture how many refinement cycles are required before the first runnable version emerges, reflecting convergence speed. Finally, the **end-to-end time cost** records the wall-clock time from pipeline start to the first executable program, characterizing practical efficiency. All results are averaged over 10 runs with up to 30 refinement cycles per run.

### 5.2 MAIN RESULTS

**MSE Results.** Table 2 shows that **Lang-PINN** achieves the lowest errors on most PDEs, despite being the only approach that must first infer PDE formulations from natural language descriptions. In contrast, *PINNacle* represents a human-expert–designed reference, where both the governing PDEs and PINN architectures are fixed in advance. Even against this strong baseline, **Lang-PINN** delivers significant improvements. For instance, errors on *KS* (1D), *Poisson-MA* (2D), and *Heat-ND* (ND) are reduced by over three orders of magnitude. Compared to agent-based baselines, the advantage is equally clear: while their errors on *KS* and *Poisson-MA* remain around $10^0$ to $10^4$, Lang-PINN reaches $10^{-3}$, demonstrating far stronger fidelity in solution quality.

**Success Rate.** Fig. 5 reports the average success rate across PDEs of different dimensionalities. **Lang-PINN** consistently delivers the highest reliability, with success exceeding 80% in 1D and 2D regimes where baselines such as RandomAgent, BayesianAgent, and PINNsAgent typically remain below 35%. Performance also remains robust in 3D, where **Lang-PINN** maintains success rates close to 75%, much higher than all baselines. **Time Overhead.** We evaluate efficiency by measuring the number of iterations required to obtain executable PINNs, with all methods capped at 50 iterations for fairness.

Table 2: Comparative performance (MSE) on 14 different PDEs (averaged over 10 runs).

| Dim | PDE | RandomAgent | BayesianAgent | PINNsAgent | SCoT | Self-Debug | Ours | PINNacle |
|-----|-----|-------------|---------------|------------|------|------------|------|----------|
| 1D | Burgers | 6.63E-02 | 8.70E-02 | 1.10E-04 | 1.40E+01 | 1.26E+01 | **6.48E-05** | 7.90E-05 |
| | Wave-C | 1.50E-01 | 1.78E-01 | 3.74E-02 | 1.28E+00 | 1.18E+00 | **2.25E-03** | 3.01E-03 |
| | KS | 1.09E+00 | 1.10E+00 | 1.09E+00 | 3.33E+00 | 2.93E+00 | **1.62E-03** | 1.04E+00 |
| 2D | Burgers-C | 2.48E-01 | 2.42E-01 | 2.93E-01 | 4.54E-01 | 4.09E-02 | **2.88E-03** | 1.09E-01 |
| | Wave-CG | 2.87E-02 | 2.11E-02 | 4.59E-02 | 2.00E+00 | 1.90E+00 | **2.52E-03** | 2.99E-02 |
| | Heat-CG | 3.96E-01 | 1.17E-01 | 9.06E-02 | 4.38E+00 | 3.81E-02 | **1.35E-03** | 8.53E-04 |
| | NS-C | 4.02E-03 | 5.12E-03 | **1.40E-05** | 5.67E-01 | 5.27E-01 | 4.05E-05 | 2.33E-05 |
| | GS | 4.28E-03 | 4.03E-03 | 3.37E+08 | 3.76E+00 | 3.35E+00 | **1.89E-03** | 4.32E-03 |
| | Heat-MS | 1.84E-02 | 7.48E-03 | 1.06E-04 | 7.10E-02 | 6.04E-03 | **2.27E-05** | 5.27E-05 |
| | Heat-VC | 3.57E-02 | 3.93E-02 | 1.43E-02 | 4.46E+00 | 4.01E-02 | **1.62E-03** | 1.76E-03 |
| | Poisson-MA | 5.87E+00 | 5.82E+00 | 3.16E+00 | 1.24E+04 | 1.07E+04 | **2.25E-03** | 1.83E+00 |
| 3D | Poisson-CG | 3.82E-02 | 2.55E-02 | 3.35E-02 | 4.17E-02 | 9.51E-03 | **1.35E-03** | 9.51E-04 |
| ND | Poisson-ND | 1.30E-04 | 4.72E-05 | 4.77E-04 | 9.93E+00 | 9.43E+00 | **8.42E-06** | 2.09E-06 |
| | Heat-ND | 2.58E-00 | **1.18E-04** | 8.57E-04 | 3.74E+00 | 3.40E-03 | 4.72E-04 | 8.52E+00 |

Our **Lang-PINN** converges in only 8 iterations on average, which is about 74% fewer than the worst baseline (31), demonstrating substantial efficiency gains. Compared to other methods such as BayesianAgent (29), PINNsAgent (21), SCoT (17), and Self-Debug (14), our **Lang-PINN** consistently reduces iteration counts, confirming that the joint design of modular code generation and feedback refinement accelerates convergence across diverse PDEs.

We also report the end-to-end time cost, measured from the start of the pipeline until runnable code is produced. As shown in Appendix 3.5, **Lang-PINN** reduces total PDE-solving time by about 21%–52% compared with all baselines.

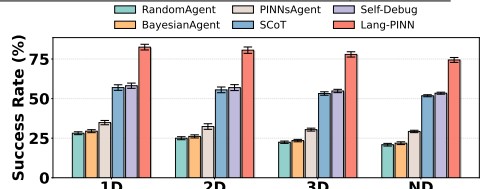

Figure 5: Comparative success rates (%) of different methods for generating executable PINNs across 1D, 2D, 3D, and ND PDEs.

### 5.3 ABLATION STUDIES

**The Impact of PDE Agent** Since Sec. 3.1 highlighted the difficulty of faithfully grounding natural-language descriptions into PDEs, we conduct an ablation study to assess the contribution of our proposed PDE Agent. Fig. 6 illustrates translation accuracy under increasing linguistic complexity. While all baselines degrade sharply from Level 1 to Level 4, our full agent consistently achieves the highest semantic consistency and maintains competitive symbolic equivalence. The gains are most evident under noisy and fragmented settings, where reasoning–canonicalization–validation steps prevent collapse and self-consistency selection stabilizes outputs. This demonstrates that the *PDE Agent* not only alleviates sensitivity to surface-form variation but also provides robust task-to-equation translation, complementing the improvements observed in MSE and executable success rate.

**The Impact of PINN Agent.** To evaluate the contribution of the *PINN Agent* in dynamically selecting architectures, we compare it with a variant where the architecture is fixed to an MLP across all PDEs, with only depth and width tuned. In contrast, the *PINN Agent* leverages PDE, prior knowledge, and history to select among different architecture families ( MLP, CNN, GNN, and Transformer). As shown in Fig. 7, dynamic selection achieves substantially lower MSEs across 14 PDEs, with the largest gains on periodic, irregular, or multi-scale problems ( KS, Poisson-MA, Heat-ND). These results highlight that the adaptive architecture selection ability of the *PINN Agent* is essential for PDE-aware architecture choice and cross-task generalization.

**The Impact of Code Agent.** To validate the Impact of the **Code Agent**, we compare its modular code generation paradigm with a monolithic generator that attempts to produce the entire code in one pass. In the monolithic setting, runtime errors are hard to localize and every correction requires regenerating the full script, resulting in fragile execution. By contrast, the Code Agent decomposes the pipeline into modules (model, loss, training loop), allowing localized correction and reuse of valid components. As shown in Fig. 8, this modular design improves the execution success rate by over 20% across PDEs, highlighting the central role of the Code Agent in ensuring executability.

**The Impact of Feedback Agent.** We next evaluate the **Feedback Agent**, focusing on how different feedback signals affect the quality of the trained PINNs. The baseline uses only error messages from failed executions to guide refinement. Our full design augments these signals with the multi-dimensional quality

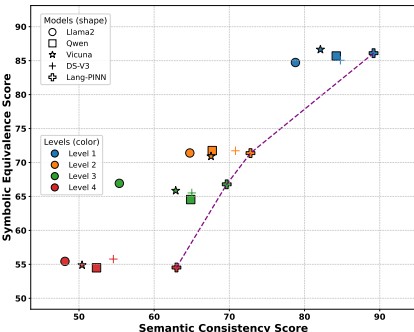

Figure 6: PDE formulation accuracy under four levels of linguistic complexity among different LLMs. Our method (**Lang-PINN**) lies on the Pareto frontier, achieving balanced improvements in both symbolic equivalence and semantic consistency scores.

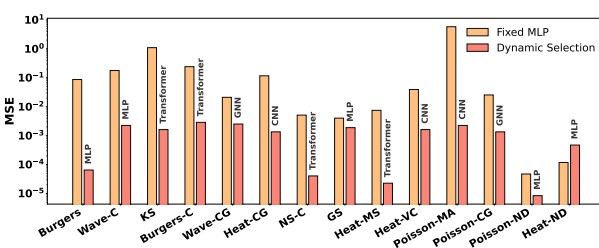

Figure 7: Ablation study of the PINN Agent: the fixed MLP variant (yellow) uses the same MLP backbone for all PDEs while the full PINN Agent (red) dynamically selects among diverse candidate architectures (e.g., MLP, CNN, and GNN). Dynamic selection consistently reduces MSE across 14 PDEs, demonstrating the effectiveness of adaptive architecture design.

metrics introduced in Sec 4.5, including loss smoothness, gradient stability, and convergence behavior. As shown in Fig. 9, the additional metrics consistently reduce MSE across PDE benchmarks, in some cases by several orders of magnitude. These results confirm that the Feedback Agent's metric-guided feedback is crucial for achieving accuracy improvements once executability has been secured by the Code Agent.

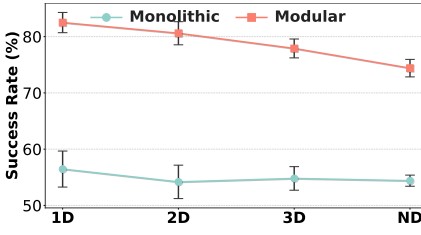

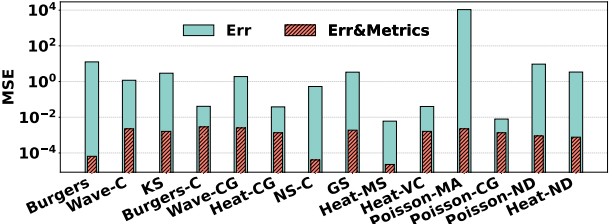

Figure 8: Ablation on the *Code Agent*: success rate (%) of monolithic vs. modular code generation.

Figure 9: Ablation on the *Feedback Agent*: MSE comparison of error-only feedback (*Err*) vs. error feedback augmented with code quality metrics (*Err&Metrics*).

Quantitatively, modular generation yields a +22% improvement in success rate, metric-guided feedback reduces mean MSE by 2.5×, and the PDE Agent improves semantic consistency by 18% on average across linguistic levels.

# 6 CONCLUSION

We introduced **Lang-PINN**, a multi-agent framework that constructs trainable physics-informed neural networks (PINNs) directly from natural-language task descriptions by integrating PDE parsing, architecture selection, modular code generation, and feedback refinement. Experiments on 14 PDEs show that **Lang-PINN** achieves lower errors, higher execution success rates, and significantly reduced time overhead compared to strong baselines, while ablations confirm the value of modular generation, feedback-driven diagnostics, and knowledge-guided design. This work highlights the potential of LLM-based agents to bridge scientific intent and executable models, with future efforts focusing on multi-physics systems, irregular geometries, and noisy real-world data.

ETHICS STATEMENT

This work does not involve human subjects, sensitive personal data, or experiments that could raise ethical concerns. The datasets used are publicly available, and no privacy or security issues are implicated. Our study focuses purely on methodological and computational aspects, and therefore we do not anticipate any direct ethical or societal risks arising from this research.

REPRODUCIBILITY STATEMENT

We have made extensive efforts to ensure the reproducibility of our results. The descriptions of the proposed models and algorithms are included in the main text, while additional implementation details, hyperparameter settings, and training procedures are provided in the appendix and supplementary material. Information about datasets and data preprocessing steps is clearly documented. To further facilitate reproducibility, we provide an anonymous repository containing the source code, experiment scripts, and configuration files in the supplementary materials.

THE USE OF LARGE LANGUAGE MODELS

In preparing this manuscript, we used LLM to refine the clarity, fluency, and readability of the English writing. The LLM was employed only for linguistic polishing and expression improvement. All scientific content, analysis, results, and conclusions were conceived, validated, and written by the authors. The authors take full responsibility for the accuracy and integrity of the scientific claims presented in this paper.

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

APPENDIX

# 1 DETAILS OF KNOWLEDGE-BASED MATCHING IN THE PINN AGENT

In Sec. 4.3 of the main text, we introduced knowledge-based matching, which aligns PDE features with architecture capabilities through a weighted similarity score. This appendix provides the detailed definitions of the PDE feature representation used on the PINN side.

## 1.1 PDE FEATURE REPRESENTATION

Each PDE $E$ is mapped to a three-dimensional feature vector

$$\phi(E) = [f_{\mathrm{per}}(E),\, f_{\mathrm{geo}}(E),\, f_{\mathrm{ms}}(E)]^{\top}, \tag{1}$$

which captures periodicity, geometry complexity, and multi-scale demand, respectively.

**Periodicity.** The degree of periodicity is quantified as

$$f_{\mathrm{per}}(E) = \frac{|\mathcal{P}(E)|}{d}, \tag{2}$$

where $d$ denotes the number of spatial dimensions and $\mathcal{P}(E)$ is the set of spatial axes with periodic boundary conditions. If all spatial directions are periodic then $|\mathcal{P}(E)| = d$ and $f_{\mathrm{per}}(E) = 1$; if none are periodic then $|\mathcal{P}(E)| = 0$ and $f_{\mathrm{per}}(E) = 0$; mixed cases lie between 0 and 1.

**Geometry complexity.** Geometry complexity combines the irregularity of the domain shape and the irregularity of the numerical discretization. We first define two scalar scores: $c_{\Omega}(E)$ for the domain and $c_{\mathrm{disc}}(E)$ for the discretization. The domain score $c_{\Omega}(E)$ is assigned as 0 for axis-aligned rectangles or boxes, 0.3 for smoothly curved domains, 0.6 for multi-component domains, and 0.9 for highly irregular or fractured geometries. The discretization score $c_{\mathrm{disc}}(E)$ is set to 0 for Cartesian grids, 0.5 for structured curvilinear grids, and 0.8 for unstructured meshes (e.g., FEM-type discretizations). We then combine these scores as

$$f_{\mathrm{geo}}(E) = \mathrm{clip}(\lambda_{\Omega}\, c_{\Omega}(E) + \lambda_{\mathrm{disc}}\, c_{\mathrm{disc}}(E),\, 0, 1)\,, \tag{3}$$

where $\lambda_{\Omega}, \lambda_{\mathrm{disc}} \geq 0$ are fixed weights satisfying $\lambda_{\Omega} + \lambda_{\mathrm{disc}} = 1$, and $\mathrm{clip}(x, 0, 1) = \min(\max(x, 0), 1)$ projects the value into $[0, 1]$. In our experiments we use $\lambda_{\Omega} = 0.6$ and $\lambda_{\mathrm{disc}} = 0.4$ and keep them fixed for all PDEs.

**Multi-scale demand.** The multi-scale demand reflects the presence of strong scale separation, nonlinear interactions, or stiff transport phenomena. We construct an intermediate score

$$\tilde{f}_{\mathrm{ms}}(E) = \alpha_1\, \mathbf{1}_{\{m(E) \geq 3\}} + \alpha_2\, \mathbf{1}_{\{\mathrm{NL}(E) = 1\}} + \alpha_3\, \log\big(1 + \mathrm{Re}(E) + \mathrm{Pe}(E)\big) + \alpha_4\, \mathbf{1}_{\{\mathrm{FR}(E) = 1\}}, \tag{4}$$

where:

- $m(E)$ is the highest derivative order in the PDE;
- $\mathrm{NL}(E) \in \{0, 1\}$ indicates whether the PDE contains nonlinear terms;
- $\mathrm{Re}(E)$ and $\mathrm{Pe}(E)$ are Reynolds and Péclet numbers when applicable (set to 0 otherwise);
- $\mathrm{FR}(E) \in \{0, 1\}$ indicates the presence of nonlocal, fractional, or integral operators;
- $\mathbf{1}_{\{\cdot\}}$ is the indicator function.

The final multi-scale feature is normalized into $[0, 1]$ by a logistic mapping,

$$f_{\mathrm{ms}}(E) = \sigma\big(\tilde{f}_{\mathrm{ms}}(E)\big) = \frac{1}{1 + \exp(-\tilde{f}_{\mathrm{ms}}(E))}. \tag{5}$$

We use the fixed weights $(\alpha_1, \alpha_2, \alpha_3, \alpha_4) = (0.8, 0.8, 0.4, 1.0)$ for all experiments.

## 1.2 PINN ARCHITECTURE CAPABILITY REPRESENTATION

To support automatic and interpretable architecture selection, we represent each PINN architecture $A$ by a three dimensional capability vector

$$\psi(A) = \big(a_{\mathrm{per}}(A),\, a_{\mathrm{geo}}(A),\, a_{\mathrm{ms}}(A)\big), \tag{6}$$

where $a_{\mathrm{per}}(A)$, $a_{\mathrm{geo}}(A)$, and $a_{\mathrm{ms}}(A)$ denote the capability of $A$ on highly periodic, geometrically complex, and strongly multi scale PDEs, respectively. All three entries lie in $[0, 1]$, with larger values indicating better

performance on the corresponding class of tasks. This vector is estimated directly from data, using controlled benchmark PDEs and a combination of absolute and relative performance measures.

We consider a collection of benchmark PDE tasks, each denoted by $E$. From the previous subsection, each task is associated with a PDE attribute vector $\phi(E) = \big(f_{\mathrm{per}}(E), f_{\mathrm{geo}}(E), f_{\mathrm{ms}}(E)\big)$, where $f_{\mathrm{per}}(E)$, $f_{\mathrm{geo}}(E)$, and $f_{\mathrm{ms}}(E)$ are scalar scores in $[0, 1]$ that quantify the periodicity, geometry complexity, and multi scale demand of $E$. Based on the largest component of $\phi(E)$, we assign each task to one of three attribute focused subsets: the high periodicity set $\mathcal{E}_{\mathrm{per}}$, the high geometry complexity set $\mathcal{E}_{\mathrm{geo}}$, and the high multi scale set $\mathcal{E}_{\mathrm{ms}}$.

For each architecture $A$ (e.g., CNN, MLP, GNN, and Transformer) and task $E$, we train a PINN and record a scalar error metric $y(A, E)$, e.g., the mean squared PDE residual or the relative $L^2$ error. The values $\{y(A, E)\}$ form an architecture–task error matrix that serves as the basis for capability estimation. At a high level, we use this matrix in three steps: 1) we first compute **normalized average scores** for each architecture within each attribute subset, 2) then derive **Bradley–Terry based relative scores** from pairwise win–loss comparisons, and 3) finally **fuse the absolute and relative scores** to obtain the capability vector $\psi(A)$.

**Absolute Capability from Normalized Error**    For a given attribute, such as periodicity, we first derive an absolute capability estimate from average error. Restricting attention to the high periodicity set $\mathcal{E}_{\mathrm{per}}$, we compute the mean error of architecture $A$ as

$$\bar{y}_{\mathrm{per}}(A) = \frac{1}{|\mathcal{E}_{\mathrm{per}}|} \sum_{E \in \mathcal{E}_{\mathrm{per}}} y(A, E), \tag{7}$$

where $|\mathcal{E}_{\mathrm{per}}|$ is the number of tasks in this subset and $\bar{y}_{\mathrm{per}}(A)$ is a scalar summarizing the overall error of $A$ on periodic tasks. We then apply min–max normalization across architectures,

$$\tilde{y}_{\mathrm{per}}(A) = \frac{\bar{y}_{\mathrm{per}}(A) - \min_{A'} \bar{y}_{\mathrm{per}}(A')}{\max_{A'} \bar{y}_{\mathrm{per}}(A') - \min_{A'} \bar{y}_{\mathrm{per}}(A') + \varepsilon}, \tag{8}$$

where $\varepsilon$ is a small constant that avoids division by zero. Finally, we convert normalized error into an absolute capability score

$$a_{\mathrm{per}}^{Abs}(A) = 1 - \tilde{y}_{\mathrm{per}}(A), \tag{9}$$

so that smaller errors correspond to larger capability values in $[0, 1]$. The same procedure applied to $\mathcal{E}_{\mathrm{geo}}$ and $\mathcal{E}_{\mathrm{ms}}$ yields the absolute geometry and multi scale capability estimates $a_{\mathrm{geo}}^{Abs}(A)$ and $a_{\mathrm{ms}}^{Abs}(A)$.

**Relative Capability via Bradley–Terry Model**    Absolute errors can be influenced by the overall difficulty of a task subset. To obtain a complementary measure that focuses on relative ordering between architectures, we employ the Bradley–Terry (BT) model (33; 20) on win–loss statistics.

Again considering the high periodicity subset $\mathcal{E}_{\mathrm{per}}$, for any pair of architectures $(A_i, A_j)$ and task $E \in \mathcal{E}_{\mathrm{per}}$, we say that $A_i$ wins over $A_j$ on $E$ if

$$y(A_i, E) < y(A_j, E). \tag{10}$$

Aggregating over all tasks in $\mathcal{E}_{\mathrm{per}}$, we count the number of wins

$$n_{ij} = \text{number of tasks in } \mathcal{E}_{\mathrm{per}} \text{ where } y(A_i, E) < y(A_j, E), \tag{11}$$

and similarly $n_{ji}$ for wins of $A_j$ over $A_i$. The collection $\{n_{ij}\}$ is treated as the observed win–loss data for periodic tasks.

The BT model introduces, for this attribute, a scalar ability parameter $\theta_{\mathrm{per}}(A)$ for each architecture $A$. This parameter is not a neural network weight, but a one dimensional statistical parameter that reflects the overall strength of $A$ on high periodicity tasks. Under the BT model, the probability that $A_i$ wins against $A_j$ in a generic periodic task is

$$p_{ij} = \Pr(A_i \succ A_j) = \frac{\exp(\theta_{\mathrm{per}}(A_i))}{\exp(\theta_{\mathrm{per}}(A_i)) + \exp(\theta_{\mathrm{per}}(A_j))}. \tag{12}$$

Given the observed win counts $n_{ij}$ and $n_{ji}$, the likelihood of these data under the model is

$$L(\theta_{\mathrm{per}}) = \prod_{i<j} p_{ij}^{n_{ij}} (1 - p_{ij})^{n_{ji}}, \tag{13}$$

where the product ranges over all unordered architecture pairs $(i, j)$ and $\theta_{\mathrm{per}}$ denotes the collection of all ability parameters. Intuitively, $L(\theta_{\mathrm{per}})$ is the probability that, if the true win probabilities were given by the BT model with parameters $\theta_{\mathrm{per}}$, one would observe exactly the win–loss counts recorded in $\{n_{ij}\}$.

We obtain the BT ability parameters by maximizing this likelihood, or equivalently the log likelihood, with respect to $\theta_{\mathrm{per}}$. This yields a set of scalar values $\theta_{\mathrm{per}}^{\star}(A)$, one for each architecture, that best explains the

observed win–loss data on periodic tasks. To convert these unnormalized abilities into a $[0, 1]$ scale, we apply min–max normalization

$$a_{\mathrm{per}}^{BT}(A) = \frac{\theta_{\mathrm{per}}^{\star}(A) - \min_{A'} \theta_{\mathrm{per}}^{\star}(A')}{\max_{A'} \theta_{\mathrm{per}}^{\star}(A') - \min_{A'} \theta_{\mathrm{per}}^{\star}(A') + \varepsilon}. \tag{14}$$

The same BT procedure applied to $\mathcal{E}_{\mathrm{geo}}$ and $\mathcal{E}_{\mathrm{ms}}$ produces relative capability scores $a_{\mathrm{geo}}^{BT}(A)$ and $a_{\mathrm{ms}}^{BT}(A)$ for geometry and multi scale attributes.

**Capability Fusion and Final Representation**  For each attribute dimension $k \in \{\mathrm{per}, \mathrm{geo}, \mathrm{ms}\}$ we now have two capability estimates: an absolute estimate $a_k^{Abs}(A)$ derived from normalized average errors, and a relative estimate $a_k^{BT}(A)$ derived from win–loss relationships. The two measures capture complementary information: absolute performance level and relative ordering across architectures. We therefore form the final capability entry along dimension $k$ by a simple linear fusion

$$a_k(A) = \omega_k \, a_k^{BT}(A) + \big(1 - \omega_k\big) \, a_k^{Abs}(A), \tag{15}$$

where $\omega_k \in [0, 1]$ is a data–driven fusion weight for attribute $k$. To determine $\omega_k$, we draw bootstrap resamples of the task subset $\mathcal{E}_k$, recompute $a_k^{BT}(A)$ and $a_k^{Abs}(A)$ on each resample, and estimate their empirical variances $\sigma_{k,\mathrm{BT}}^2$ and $\sigma_{k,\mathrm{Abs}}^2$. We then set

$$\omega_k = \frac{\sigma_{k,\mathrm{Abs}}^2}{\sigma_{k,\mathrm{Abs}}^2 + \sigma_{k,\mathrm{BT}}^2}, \tag{16}$$

so that the estimator with smaller variance (more stable across resamples) receives a larger effective weight.

The final architecture capability vector

$$\psi(A) = \big(a_{\mathrm{per}}(A), \, a_{\mathrm{geo}}(A), \, a_{\mathrm{ms}}(A)\big) \tag{17}$$

is stored in the knowledge base and later matched against PDE attribute vectors $\phi(E)$ for new tasks. This representation allows the system to reason about architecture–PDE alignment in a quantitative and interpretable manner.

## 2   FEEDBACK AGENT QUALITY METRICS

The validation score produced by the *Feedback Agent* agent aggregates four normalized metrics, each designed to capture a complementary aspect of code quality. Below we detail the first three metrics; the robustness metric is described separately.

**(i) Convergence efficiency.**  Convergence efficiency measures how quickly a model reaches a stable solution. We define it based on the number of training steps required for the loss to fall below a pre-specified tolerance $\tau$:

$$T_{\mathrm{conv}} = \min\{t \mid L_t \leq \tau\}, \quad m_{\mathrm{conv}} = \frac{1}{T_{\mathrm{conv}}}, \tag{18}$$

where $L_t$ denotes the training loss at iteration $t$. A smaller $T_{\mathrm{conv}}$ leads to a higher convergence score. For comparability across models, we normalize the score using the range of convergence steps observed in the search space:

$$\hat{m}_{\mathrm{conv}} = \frac{T_{\max} - T_{\mathrm{conv}}}{T_{\max} - T_{\min}}, \tag{19}$$

where $T_{\min}$ and $T_{\max}$ denote, respectively, the fastest and slowest convergence times among all candidates. This normalization ensures $\hat{m}_{\mathrm{conv}} \in [0, 1]$, with higher values indicating more efficient convergence.

**(ii) Predictive accuracy.**  Accuracy is assessed by the discrepancy between the model output and the governing PDE. Specifically, we compute the mean squared error (MSE) of the PDE residual over the training domain:

$$m_{\mathrm{acc}} = -\mathrm{MSE}\big(\mathcal{N}_\theta, E\big), \tag{20}$$

where $\mathcal{N}_\theta$ denotes the physics-informed neural network (PINN) parameterized by $\theta$, and $E$ represents the target PDE operator. The negative sign ensures that lower residual error corresponds to a higher accuracy score.

**(iii) Model complexity.**  Complexity reflects the resource demand of the model. We quantify it by the number of trainable parameters (or equivalently the computational cost in FLOPs), normalized with respect to the maximum within the search space:

$$m_{\mathrm{comp}} = \frac{\#\mathrm{Params}(\mathcal{N}_\theta)}{\max \#\mathrm{Params}}, \tag{21}$$

where $\#\mathrm{Params}(\mathcal{N}_\theta)$ is the parameter count of the candidate PINN and $\max \#\mathrm{Params}$ is the maximum parameter count among all models considered. A lower value of $m_{\mathrm{comp}}$ indicates a more compact architecture.

**(iv) Robustness.** We quantify robustness by combining two complementary indicators. The first indicator, *loss smoothness*, measures the stability of the training trajectory. Intuitively, when the loss fluctuates strongly across iterations, the optimization process is less reliable. We capture this by computing the normalized variation of the loss:

$$m_{\text{smooth}} = 1 - \frac{\text{Std}(\Delta L_t)}{\text{Mean}(L_t)}, \quad \Delta L_t = L_t - L_{t-1}, \tag{22}$$

where $L_t$ denotes the training loss at iteration $t$, and $\Delta L_t$ is the difference between consecutive iterations. A higher value of $m_{\text{smooth}}$ indicates a smoother and more stable training curve.

The second indicator, *gradient health*, evaluates whether the gradient magnitude remains within a reasonable range, avoiding both vanishing and exploding gradients. Specifically,

$$m_{\text{grad}} = \begin{cases} 1, & \epsilon \le \dfrac{\|\nabla_\theta L\|}{d} \le \kappa, \\ 0, & \text{otherwise}, \end{cases} \tag{23}$$

where $\nabla_\theta L$ is the gradient of the loss with respect to the parameters, $d$ is the number of parameters, and $\epsilon, \kappa > 0$ are user-defined thresholds specifying the acceptable lower and upper bounds of the normalized gradient magnitude.

Finally, we define the robustness score as a convex combination of the two indicators:

$$m_{\text{rob}} = \alpha\, m_{\text{smooth}} + (1 - \alpha)\, m_{\text{grad}}, \tag{24}$$

where $\alpha \in [0, 1]$ is a weighting factor that balances the contributions of loss smoothness and gradient health. This formulation ensures that robustness reflects both stable optimization dynamics and well-conditioned gradients.

The overall validation score is defined as a weighted combination of the four normalized metrics:

$$S(C) = w_1\, \hat{m}_{\text{conv}} + w_2\, \hat{m}_{\text{acc}} + w_3\, \hat{m}_{\text{comp}} + w_4\, \hat{m}_{\text{rob}}, \tag{25}$$

where $w_1, w_2, w_3, w_4 \ge 0$ are user-specified weights that control the relative importance of convergence efficiency, predictive accuracy, model complexity, and robustness, respectively. By tuning the weights, one can emphasize different aspects of model quality depending on the application.

## 3 EXTENDED RESULTS

### 3.1 MSE AND SUCCESS RATE ACROSS PDE BENCHMARKS

For completeness, we report the full experimental results across all 14 PDE benchmarks. Table 1 presents the mean squared error (MSE) together with standard deviations, complementing the aggregated results in the main text. Table 2 provides per-PDE success rates (%) averaged over 10 runs, offering a more fine-grained view of performance across different equations and dimensions.

Table 1: Comparative performance (MSE) of **Lang-PINN** and baseline approaches on 14 different PDEs. Results are averaged over 10 runs.

| PDEs | RandomAgent | BayesianAgent | PINNsAgent | PINNacle | SCoT | Self-Debug | Ours |
|------|-------------|---------------|------------|----------|------|------------|------|
| **1D** | | | | | | | |
| Burgers | 6.63E-02 (±1.10E-01) | 8.70E-02 (±6.51E-03) | 1.10E-04 (±7.76E-05) | 7.90E-05 | 1.40E+01 (±1.06E+00) | 1.26E+01 (±9.54E-01) | 6.48E-05 (±9.00E-05) |
| Wave-C | 1.50E-01 (±1.46E-01) | 1.78E-01 (±3.84E-02) | 3.74E-02 (±4.32E-02) | 3.01E-03 | 1.28E+00 (±6.21E-02) | 1.18E+00 (±5.72E-02) | 2.25E-03 (±1.80E-04) |
| KS | 1.09E+00 (±3.58E-02) | 1.10E+00 (±2.55E-03) | 1.09E+00 (±3.20E-02) | 1.04E+00 | 3.33E+00 (±7.80E-02) | 2.93E+00 (±6.86E-02) | 1.62E-03 (±1.35E-04) |
| **2D** | | | | | | | |
| Burgers-C | 2.48E-01 (±4.04E-03) | 2.42E-01 (±8.96E-03) | 2.93E-01 (±2.43E-02) | 1.09E-01 | 4.54E-01 (±5.57E-02) | 4.09E-02 (±5.01E-03) | 2.88E-03 (±2.25E-04) |
| Wave-CG | 2.87E-02 (±4.98E-04) | 2.11E-02 (±1.12E-02) | 4.59E-02 (±1.68E-02) | 2.99E-02 | 2.00E+00 (±1.62E-01) | 1.90E+00 (±1.54E-01) | 2.52E-03 (±1.62E-04) |
| Heat-CG | 3.96E-01 (±3.22E-01) | 1.17E-01 (±3.24E-02) | 9.06E-02 (±2.69E-01) | 8.53E-04 | 4.38E+00 (±3.48E-01) | 3.81E-02 (±3.03E-03) | 1.35E-03 (±9.00E-05) |
| NS-C | 4.02E-03 (±5.93E-03) | 5.12E-03 (±1.33E-03) | 1.40E-05 (±1.12E-05) | 2.33E-05 | 5.67E-01 (±6.28E-02) | 5.27E-01 (±5.84E-02) | 4.05E-05 (±4.50E-05) |
| GS | 4.28E-03 (±2.23E-05) | 4.03E-03 (±4.47E-04) | 3.37E+08 (±1.01E+09) | 4.32E-03 | 3.76E+00 (±5.27E-02) | 3.35E+00 (±4.69E-02) | 1.89E-03 (±1.44E-04) |
| Heat-MS | 1.84E-02 (±1.18E-02) | 7.48E-03 (±3.81E-03) | 1.06E-04 (±1.86E-04) | 5.27E-05 | 7.10E-02 (±3.05E-03) | 6.04E-03 (±2.59E-04) | 2.27E-05 (±7.20E-05) |
| Heat-VC | 3.57E-02 (±8.72E-03) | 3.93E-02 (±2.17E-03) | 1.43E-02 (±1.77E-02) | 1.76E-03 | 4.46E+00 (±1.05E+00) | 4.01E-02 (±9.45E-03) | 1.62E-03 (±1.08E-04) |
| Poisson-MA | 5.87E+00 (±1.17E+00) | 5.82E+00 (±2.30E+00) | 3.16E+00 (±9.92E-01) | 1.83E+00 | 1.24E+04 (±5.71E+03) | 1.07E+04 (±4.91E+03) | 2.25E-03 (±1.35E-04) |
| **3D** | | | | | | | |
| Poisson-CG | 3.82E-02 (±2.15E-02) | 2.55E-02 (±5.65E-03) | 3.35E-02 (±2.18E-02) | 9.51E-04 | 4.17E-02±3.77e-03 | 9.51E-03±1.35e-03 | 1.35E-03 (±9.00E-05) |
| **ND** | | | | | | | |
| Poisson-ND | 1.30E-04 (±2.78E-04) | 4.72E-05 (±2.76E-06) | 4.77E-04 (±3.21E-05) | 2.09E-06 | 9.93E+00 (±6.51E-03) | 9.43E+00 (±6.18E-03) | 842.00E-06 (±5.17E-07) |
| Heat-ND | 2.58E-00 (±9.87E-02) | 1.18E-04 (±8.92E-06) | 8.57E-04 (±1.31E-06) | 8.52E+00 | 3.74E+00 (±3.29E-01) | 3.40E-03 (±2.99E-04) | 4.72E-04 (±6.30E-05) |

These results serve as a detailed supplement to the main comparisons: our method consistently achieves the lowest average errors with significantly reduced variance, and obtains higher success rates across nearly all

PDEs. In particular, **Lang-PINN** improves code executability and training stability even for challenging high-dimensional and chaotic cases, reinforcing the conclusions drawn in the main paper.

Table 2: Success rate (%) of **Lang-PINN** and baseline approaches on 14 different PDEs. Results are averaged over 10 runs.

| PDEs | RandomAgent | PINNsAgent | PINNacle | SCoT | Self-Debug | Ours |
|------|-------------|------------|----------|------|------------|------|
| **1D** | | | | | | |
| Burgers | 29.7% | 36.2% | 38.9% | 58.6% | 59.7% | 84.3% |
| Wave-C | 28.5% | 34.8% | 37.2% | 57.2% | 58.3% | 80.7% |
| KS | 27.9% | 33.5% | 35.9% | 55.1% | 56.4% | 82.5% |
| **2D** | | | | | | |
| Burgers-C | 26.1% | 33.4% | 36.2% | 56.3% | 58.0% | 81.1% |
| Wave-CG | 25.4% | 31.2% | 34.0% | 54.9% | 56.1% | 77.4% |
| Heat-CG | 25.1% | 32.6% | 35.1% | 55.7% | 57.0% | 81.6% |
| NS-C | 26.3% | 34.1% | 36.8% | 57.1% | 58.9% | 83.3% |
| GS | 24.9% | 30.7% | 33.2% | 53.8% | 55.0% | 78.8% |
| Heat-MS | 26.8% | 35.0% | 37.6% | 58.4% | 59.6% | 82.7% |
| Heat-VC | 25.6% | 32.0% | 34.5% | 55.2% | 56.8% | 80.5% |
| Poisson-MA | 23.7% | 29.8% | 32.7% | 52.7% | 54.1% | 79.2% |
| **3D** | | | | | | |
| Poisson-CG | 22.9% | 30.4% | 33.5% | 53.2% | 54.8% | 77.9% |
| **ND** | | | | | | |
| Poisson-ND | 21.7% | 28.9% | 31.7% | 51.5% | 53.1% | 73.3% |
| Heat-ND | 20.9% | 29.6% | 32.4% | 52.1% | 53.7% | 75.5% |

## 3.2 Effectiveness of Semantic–Symbolic PDE Verification

To assess the effectiveness of the proposed semantic–symbolic verification, we perform an evaluation on a held-out collection of PDE tasks. For each task, the LLM generates multiple candidate PDEs. These candidates are grouped into five quality categories based on symbolic correctness (operators, coefficients, and BC/IC structure). For every generated PDE, we compute its semantic consistency score by comparing the model's natural-language explanation of the equation with the original task description, ensuring that all governing components are aligned.

To measure how well this score reflects actual PDE quality, each candidate PDE is passed through the remaining agents and used to train a PINN, from which we obtain the final mean-squared error. The results in Table 3 show a clear trend: **PDEs with higher semantic consistency yield lower PINN error** (equivalently, larger $-\log_{10}$MSE). Across all five quality groups, the semantic score exhibits a strong monotonic relationship with downstream accuracy, with a Pearson correlation of $r = 0.88$. These results indicate that semantic–symbolic validation provides a reliable, data-supported proxy for identifying missing constraints and assessing PDE correctness before PINN training.

## 3.3 Effectiveness of Semantic–Symbolic PDE Verification

To assess whether our semantic–symbolic verification reliably reflects PDE quality, we generate multiple PDE candidates for several benchmark tasks and group them into five perturbation classes: C1 (perfect PDE), C2 (notation-level variation), C3 (coefficient error), C4 (missing or incorrect terms), and C5 (structural error or hallucination). For each candidate, we compute the semantic-consistency score and then train a PINN using that PDE to obtain the final $-\log_{10}!$ MSE.

Table 3 shows a clear monotonic trend across all four PDEs: as the perturbation becomes more severe, the semantic score decreases and the resulting PINN error increases. Across the entire evaluation set, the semantic score exhibits a strong negative correlation with the final training error (Pearson correlation $r = 0.88$). These results confirm that the semantic–symbolic metric provides a reliable, data-supported proxy for detecting missing or incorrect constraints prior to PINN training.

Table 3: Semantic consistency score and PDE MSE loss across five quality levels of LLM-generated PDEs. Categories C1–C5 correspond respectively to: exactly correct PDEs, notation-only variations, coefficient errors, missing/incorrect terms, and structural hallucinations.

| PDE Category | Semantic Consistency Score / $-\text{Log}_{10}$ MSE | | | |
| --- | --- | --- | --- | --- |
| | **Burgers** | **Heat-MS** | **Wave-C** | **KS** |
| C1 (exactly correct) | 1.00 / 4.1884 | 1.00 / 4.6445 | 1.00 / 2.6492 | 1.00 / 2.6055 |
| C2 | 0.91 / 3.8915 | 0.89 / 4.3101 | 0.92 / 2.4302 | 0.86 / 2.2156 |
| C3 | 0.71 / 0.6341 | 0.75 / 0.7478 | 0.69 / -0.5861 | 0.70 / -0.8446 |
| C4 | 0.51 / -0.0546 | 0.59 / -0.5053 | 0.45 / -1.5269 | 0.53 / -1.1914 |
| C5 | 0.28 / -0.6618 | 0.23 / -0.8240 | 0.14 / -2.1447 | 0.22 / -2.1026 |
| **Pearson correlation** | **0.88** | | | |

### 3.4 SEMANTIC CONSISTENCY THRESHOLDING

To evaluate the reliability of the semantic-consistency metric, we adopt an LLM-as-a-judge procedure. For any pair of textual task descriptions $d_i$ and $d_j$, the judge model is prompted with both descriptions and returns a similarity score $s \in [0, 1]$, where larger values indicate stronger semantic alignment.

For threshold calibration, we construct a benchmark containing 200 equivalent pairs and 200 non-equivalent pairs. Each equivalent pair is formed by taking two independently re-expressed descriptions of the same underlying PDE from the Task2PDE dataset. Each non-equivalent pair is formed by taking two descriptions drawn from two different PDEs from Task2PDE dataset.

The score distribution in Fig 1 shows a clear separation. Non-equivalent pairs concentrate below $0.70$, while equivalent pairs mostly lie above $0.75$, with only a narrow overlap between the two. Based on this separation, we adopt a conservative threshold of $0.80$, which retains virtually all equivalent pairs while rejecting nearly all non-equivalent ones. This threshold is used throughout Section 4 to validate LLM-generated PDE formulations.

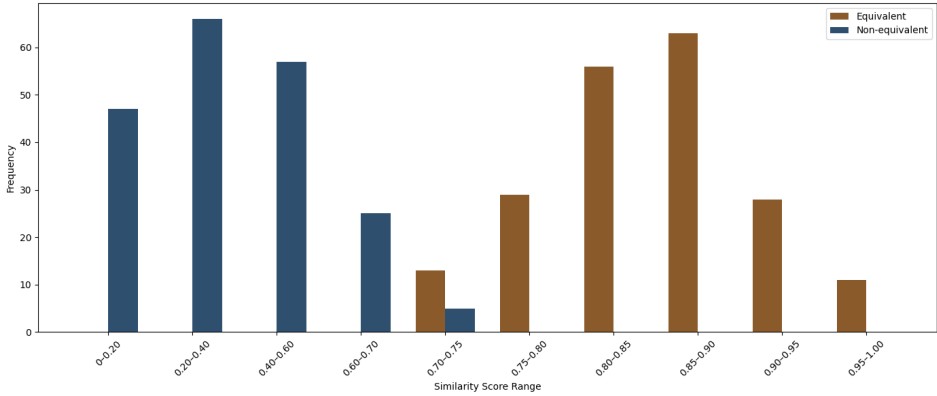

Figure 1: Similarity score distribution of 200 equivalent vs. 200 non-equivalent pairs.

### 3.5 END-TO-END WALL-CLOCK TIME EVALUATION

We assess the end-to-end runtime of the full pipeline, measured from pipeline start to until runnable code is produced. In our implementation, the LLM component is accessed through an API (e.g., DeepSeek-V3), and all model computation runs on the provider's backend. As device-level memory consumption of the LLM is therefore not observable, we focus on **wall-clock time**, which captures the total latency introduced by both LLM inference and subsequent refinement steps.

Each method is executed 10 times under a unified protocol, allowing up to 30 refinement cycles per run. Measured from pipeline start to until runnable code is produced. As shown in Table 4 **Lang-PINN achieves a 21%–52% reduction in total PDE-solving time relative to all baselines**, demonstrating that the proposed multi-agent workflow improves overall efficiency while maintaining solution quality.

Table 4: End-to-end wall-clock time, measured from pipeline start until runnable code is produced.

| Method | RandomAgent | BayesianAgent | PINNsAgent | SCoT | Self-Debug | Lang-PINN |
|---|---|---|---|---|---|---|
| Avg. Time (s) | 413.5 | 391.2 | 312.8 | 291.1 | 246.4 | **199.7** |

## 3.6 THE IMPACT OF THE LLM BACKBONE.

To evaluate how the backbone language model affects Lang-PINN, we run the complete multi-agent workflow with three different LLMs: DeepSeek-V3, Qwen2, and LLaMA2-Chat. All experiments use identical decoding settings and fixed prompt templates. Table 5 shows that although DeepSeek-V3 yields the lowest absolute MSE, Lang-PINN with weaker backbones such as Qwen2 and LLaMA2-Chat still achieves lower error than all baselines that rely on DeepSeek-V3. This indicates that the performance improvement primarily comes from the multi-agent framework, including verification and fallback across the PDE, PINN, Code, and Feedback Agents, rather than from the strength of any specific LLM. A stronger model further improves intermediate reasoning, but the relative gain provided by Lang-PINN remains consistent across all LLM families, showing that the framework generalizes well and is not dependent on a single proprietary backbone.

Table 5: Comparison of PINN MSE across different LLM backbones.

| PDE | Random | Bayesian | SCoT | Self-Debug | Lang-PINN (DeepSeek-V3) | Lang-PINN (Qwen2) | Lang-PINN (LLaMA2) |
|---|---|---|---|---|---|---|---|
| KS | 1.09E+00 | 1.10E+00 | 3.33E+00 | 2.93E+00 | 1.62E-03 | 1.95E-03 | 2.71E-03 |
| NS-C (2D) | 4.02E-03 | 5.12E-03 | 5.67E-03 | 5.27E-01 | 4.05E-05 | 5.47E-05 | 6.88E-05 |
| Poisson-MA | 5.87E+00 | 5.82E+00 | 1.24E+00 | 1.07E+00 | 2.25E-03 | 2.83E-03 | 3.22E-03 |
| GS (2D) | 4.28E-03 | 4.03E-03 | 5.35E-03 | 5.35E-03 | 1.89E-03 | 2.42E-03 | 3.16E-03 |

## 4 TASK2PDE DATASET

To rigorously evaluate the ability of language models to map natural-language task descriptions into formal PDE specifications, we construct the **Task2PDE** dataset. The dataset is derived from eight representative PDE families selected from the PINNacle benchmark (7), spanning different spatial dimensions:

- **1D:** Burgers', Wave–C, Kuramoto–Sivashinsky (KS);

- **2D:** Heat–MS, Poisson–MA, incompressible Navier–Stokes (NS–C);

- **3D:** Poisson–CG;

- **High-dimensional ND:** Heat–ND.

For each PDE family, we construct 50 distinct task descriptions under four difficulty levels, yielding a total of $8 \times 4 \times 50 = 1600$ samples. Each sample is paired with its ground-truth PDE formulation, including operators, coefficients, boundary/initial conditions, and domain specification. This ensures that every natural-language description corresponds uniquely to one PDE instance, enabling systematic evaluation of semantic-to-symbol grounding.

### 4.1 LINGUISTIC COMPLEXITY LEVELS IN TASK2PDE

For each forward problem in Task2PDE, we construct four task descriptions that all refer to the *same* underlying PDE, but differ in how realistically they reflect human-written text. These four levels are designed to mimic how researchers actually describe PDE-based problems in practice, from clean paper-style statements to noisy, ambiguous, and disorganized notes. In the examples below, we highlight in **bold** the phrases that introduce the level-specific difficulty, and we render each level in a different color for clarity.

#### 4.1.1 LEVEL 1: CLEAN AND EXPLICIT DESCRIPTION.

Level 1 corresponds to a concise and well-structured description, as one would expect in a paper or textbook. The PDE, domain, and boundary/initial conditions are stated explicitly, with no irrelevant or ambiguous information, so the mapping from text to PDE is essentially direct.

---

**Example of Level 1**

We consider one-dimensional heat diffusion in a rod of length $L = 1$ with constant thermal conductivity $\kappa = 0.01$. The temperature $u(x,t)$ satisfies

$$\partial_t u(x,t) = \kappa \, \partial_{xx} u(x,t), \quad x \in [0,1], \ t > 0.$$

Boundary conditions are $u(0,t) = 0$ and $u(1,t) = 1$ for all $t > 0$, and the initial condition is $u(x,0) = \sin(\pi x)$.

---

In this case, a correct Task2PDE model should recover exactly the ground-truth PDE and its boundary and initial conditions.

### 4.1.2 Level 2: Irrelevant but realistic side information.

Level 2 keeps the same PDE as Level 1, but mixes in realistically irrelevant details that researchers often include in emails or lab notes, such as comments about the experimental environment or personal impressions. These phrases should not affect the governing equation, yet they increase the risk that an LLM mistakes them for physical conditions.

---

**Example of Level 2**

We are again simulating heat diffusion in a metal rod of length 1 with constant conductivity $\kappa = 0.01$ using the standard heat equation on $[0,1]$. **The lab was quite cold in the morning and the left end of the setup felt a bit colder when I touched it, but this is just due to the room air and is not part of the mathematical model.** In the simulation, we still impose $u(0,t) = 0$ and $u(1,t) = 1$ for all $t > 0$, and use $u(x,0) = \sin(\pi x)$ as the initial condition.

---

The bolded sentences are natural in real experimental notes but do not belong to the PDE constraints. A robust system should ignore these side comments and recover the same PDE and BC/IC as in Level 1; a less robust system may, for example, turn the qualitative remark about "felt a bit colder" into a spurious time-dependent boundary condition.

### 4.1.3 Level 3: Ambiguous wording and underspecified terminology.

Level 3 models the situation where the researcher assumes that the reader shares the experimental context, and therefore uses shorthand or ambiguous phrases without fully specifying what they refer to. These expressions are understandable to humans who know the setup, but they can blur the distinction between measurement noise and true physical variation.

---

**Example of Level 3**

We revisit the same 1D heat conduction setup on $[0,1]$ with a homogeneous material. **At the left end, the temperature reading tends to drift over time because the sensor is not very stable, but in the actual experiment the boundary itself is kept at a fixed $0°$C throughout the run.** The right end is maintained at $1°$C, and we use thermal conductivity $\kappa = 0.01$ with the same sine-shaped initial profile as in our standard diffusion case.

---

Here, the phrase **"the temperature reading tends to drift over time"** refers to sensor drift rather than a time-varying boundary condition, while the follow-up sentence clarifies that the boundary temperature is fixed. A correct Task2PDE model should resolve this ambiguity and still output $u(0,t) = 0$ and $u(1,t) = 1$ as in Level 1. A misinterpreting model may instead treat the drift as a genuine time-dependent boundary, leading to an incorrect PDE–BC pairing.

### 4.1.4 Level 4: Disorganized, out-of-order description.

Level 4 reflects free-form lab notebook or chat-style descriptions, where the researcher writes conditions in the order they occur to mind rather than in a structured way. All information needed to reconstruct the same PDE as in Level 1 is present, but it is scattered, partially repeated, and appears in a non-linear order, often mixing preliminary and final settings.

---

**Example of Level 4**

For this batch of runs we use the same basic heat diffusion setup as before on a rod from $x = 0$ to $x = 1$. **Initially we tried several values for the thermal conductivity, like $\kappa = 0.005$ and $\kappa = 0.02$, but in the final configuration we fixed it at $\kappa = 0.01$.** The initial temperature profile is the sine-shaped one from our earlier tests. **The right end is kept at temperature $1$ during the whole experiment. At the left end, even though the hardware was moved during calibration and the sensor readings jumped a bit, the boundary itself was maintained at $0$ for the entire run.** The evolution of $u(x, t)$ is still governed by the standard heat equation.

---

The highlighted phrases illustrate typical disorganization: multiple candidate values of $\kappa$ appear before the final choice, and the left boundary condition is embedded in comments about hardware motion and sensor jumps. Human readers can usually infer that the true settings are $\kappa = 0.01$, $u(0, t) = 0$, and $u(1, t) = 1$. An LLM that fails to integrate these scattered cues may instead latch onto a preliminary $\kappa$ or ignore the final clarification about the boundary, thus producing an incorrect PDE or boundary conditions even though all necessary information is present in the text.

## 5  PROMPT DESIGN DETAILS OF ALL AGENTS

In this section, we provide the detailed prompts of our all agents, including PDE Agent, PINN Agent, Code Agent, and Feedback Agent.

### 5.1  PDE AGENT

The PDE Agent converts natural-language physical descriptions into multiple plausible governing PDE candidates, expressed in normalized residual form with structured metadata (variables, parameters, domains, and IC/BC).

#### 5.1.1  SYSTEM PROMPT

**PDE Agent — System Prompt**

```
You are the PDE Agent in a multi-agent PDE-to-PINN system.
Your task is to infer plausible governing PDEs from natural-language
descriptions of physical systems, without using any ground-truth
    labels.

You must:
- parse the given physical description d;
- reason about the underlying operators, propagation mechanisms,
  and boundary behavior;
- generate K independent reasoning trajectories {T1, ..., TK};
- in each trajectory, infer exactly ONE plausible PDE candidate Ei;
- write the PDE in normalized residual form F(u, x, t; theta) = 0;
- list variables, parameters, and the space-time domain;
- extract initial and boundary conditions when they are implied;
- provide a 24 sentence chain-of-thought explanation for each Ei;
- return all trajectories in a single JSON object with the schema:

{
  "trajectories": [
    {
      "id": "Ti",
      "reasoning": "<natural-language explanation>",
      "residual_form": "<PDE residual = 0>",
      "variables": "<state and coordinates>",
      "parameters": "<physical parameters>",
      "domain": "<space-time domain>",
      "ic": "<initial condition>",
      "bc": "<boundary condition(s)>"
    },
```

```
      ...
    ]
  }

  Follow the JSON schema exactly and do NOT output any text outside
      JSON.
```

### 5.1.2 USER PROMPT

**PDE Agent — User Prompt (1D Wave Example)**

```
Given the following physical description d, generate K = 5
    independent
reasoning trajectories T1T5. For each trajectory, infer one
    plausible PDE
candidate in residual form F(u, x, t; theta) = 0, following the
    required JSON
schema.

[Description d]
Consider a long, taut string fixed at both ends, positioned
    horizontally in a
controlled laboratory environment. Initially, the string is
    displaced from its
rest position according to a known spatial profile, and it may also
    possess an
initial velocity distribution along its length. Throughout the
    experiment, the
endpoints remain immobile, ensuring that disturbances can only
    travel along the
string. The surrounding air resistance is assumed negligible, so
    only the
inherent properties of the string influence the motion. As time
    advances, the
displacement of each point along the string varies due to the
    propagation and
reflection of waves, potentially creating complex oscillatory
    patterns. What
mathematical equation should be formulated to describe how the
    vertical
displacement of the string at any position changes as a function of
    both time
and spatial coordinate, given the initial shape, initial velocity,
    and fixed
boundary conditions?

Use this description d to construct 5 reasoning trajectories and
    output the
final result strictly in JSON.
```

### 5.1.3 ASSISTANT OUTPUT

**PDE Agent — Assistant Output (1D Wave)**

**(a) JSON reasoning trajectories:**

```
{
  "trajectories": [
    {
      "id": "T1",
      "reasoning": "The description is a textbook setup of a
      long, taut string with small transverse displacements,
      fixed at both ends, and negligible air resistance, so the
      dominant physics is elastic wave propagation along one
      spatial dimension. For such a uniform string, the balance
      of transverse forces and inertia leads to the 1D linear
      wave equation with a constant wave speed determined by
      string tension and linear density. The fixed endpoints
      impose zero displacement at both ends of the interval in
      space. The initial shape and initial velocity translate
      directly into initial conditions for u and its first time
      derivative.",
      "residual_form": "F(u,x,t;theta) = u_tt(x,t) - c^2
      u_xx(x,t) = 0",
      "variables": "u(x,t): vertical displacement of the
      string; x: position along the string; t: time.",
      "parameters": "theta = {c, L}, where c > 0 is the
      constant wave speed (c^2 = T/ with tension T and linear
      density ) and L is the string length.",
      "domain": "0 < x < L, t > 0.",
      "ic": "u(x,0) = f(x) (given initial shape), u_t(x,0) =
      g(x) (given initial velocity profile).",
      "bc": "u(0,t) = 0, u(L,t) = 0 for all t  0 (fixed
      endpoints)."
    },
    {
      "id": "T2",
      "reasoning": "Instead of introducing the wave speed
      explicitly, we can write the governing law directly from
      Newtons second law for a small string element. The net
      transverse force is given by the difference of tensions
      at its ends, which in the continuous limit produces a
      term proportional to the second spatial derivative of
      displacement. Balancing this force with the inertial term
      involving the linear mass density yields a wave equation
      with tension and density as the primary parameters.",
      "residual_form": "F(u,x,t;theta) =  u_tt(x,t) - T
      u_xx(x,t) = 0",
      "variables": "u(x,t): vertical displacement; x:
      coordinate along the string from 0 to L; t: time.",
      "parameters": "theta = {T, , L}, where T > 0 is the
      constant string tension and  > 0 is the constant linear
      mass density.",
      "domain": "0 < x < L, t > 0.",
      "ic": "u(x,0) = f(x), u_t(x,0) = g(x), where f and g
      are the prescribed initial displacement and velocity
      distributions.",
      "bc": "u(0,t) = 0, u(L,t) = 0 for all t  0 (both ends
      clamped and immobile)."
    },
    ......
    }
  ]
}
```

**(b) Canonical PDE after consensus voting (XML form):**

```
<root>
  <parsed_pde>u_tt - c^2*u_xx = 0</parsed_pde>
  <variables>u = u(x,t); c = wave speed; x in [0,L]; t >=
    0</variables>
  <domain>x in [0,L], t >= 0</domain>
  <initial_condition>
    u(x,0) = u0(x); u_t(x,0) = v0(x)
  </initial_condition>
  <boundary_condition>
    u(0,t) = 0; u(L,t) = 0 for all t >= 0
  </boundary_condition>
</root>
```

## 5.2 PINN AGENT

The PINN Agent selects the most appropriate PINN architecture for a given PDE, using a combination of physical reasoning, history reuse, knowledge-base capabilities, and feature–capability matching.

### 5.2.1 SYSTEM PROMPT

**PINN Agent — System Prompt**

```
You are the PINN Architecture Agent in a multi-agent PDE-to-PINN
   system.
Your role is to determine the most suitable architecture for a
   given PDE by
combining physical reasoning, history reuse (H), knowledge-base
   capability
inference (K), and featurecapability matching.

Follow this reasoning workflow:
1. Parse the PDE-XML to identify operators, nonlinearity, IC/BC,
   geometry,
   periodicity, and multiscale characteristics.
2. Construct the PDE feature vector phi(E) = [f_per, f_geo, f_ms]
   [0,1]^3.
3. Query the history cache H:
   - Compute semantic similarity between the current PDE and past
   tasks.
   - If a close match exists, reuse its architecture and
   hyperparameters.
4. Query knowledge base K:
   - Infer capability vectors psi(A)  [0.1,0.9]^3 for each candidate
     architecture A  {MLP, CNN, GNN, Transformer}.
   - Ground these capabilities in Ks empirical performance and
    known inductive
     biases of these architecture families.
5. Perform featurecapability matching with weighted cosine
    similarity:
      S(A|E) = <W  phi(E), psi(A)> /
               (||W  phi(E)|| * ||psi(A)||)
6. Output a JSON object containing history_match, pde_features,
   candidate_architectures, capability_vectors, weights, scores,
   selected, config, backup, and rationale.

Do NOT output text outside JSON.
```

### 5.2.2 USER PROMPT

---

**PINN Agent — User Prompt**

```
Given the PDE-XML, history cache H, and knowledge base K, select
    the optimal
architecture for the task.

Procedure:
1) History Reuse:
   - Compare PDE structure + IC/BC + features with entries in H.
   - If similar, directly reuse the stored architecture.

2) FeatureCapability Matching:
   - Compute phi(E)  [0,1]^3 for periodicity, geometry complexity,
    and
     multiscale demand.
   - Infer psi(A) for each architecture in:
       ["MLP", "CNN", "GNN", "Transformer"].
   - Use weights W and compute S(A|E) for each candidate.
   - Rank and select the best architecture.

Return JSON only:
{
  "history_match": {...},
  "pde_features": {...},
  "candidate_architectures": [...],
  "capability_vectors": {...},
  "weights": {...},
  "scores": {...},
  "selected": "...",
  "config": {...},
  "backup": [...],
  "rationale": "..."
}

# PDE-XML:
{pde_xml}
```

---

### 5.2.3 ASSISTANT OUTPUT

---

**Assistant Output**

**Example JSON Output:**

---

```
{
  "history_match": {
     "matched": true,
     "source_task_id": "wave_1d_fixed_ends",
     "reason": "The PDE-XML describes a 1D linear wave
     equation u_tt - c^2 u_xx = 0 on x in [0,L] with fixed-end
     Dirichlet boundary conditions and prescribed initial
     displacement and velocity, which closely matches a
     previous wave-string task in H."
  },

  "pde_features": {
    "f_per": 0.1,
    "f_geo": 0.2,
    "f_ms": 0.8
  },

  "candidate_architectures": [
    "MLP", "CNN", "GNN", "Transformer"
  ],

  "capability_vectors": {
    "MLP":         [0.2, 0.2, 0.5],
    "CNN":         [0.1, 0.6, 0.6],
    "GNN":         [0.1, 0.7, 0.4],
    "Transformer": [0.3, 0.4, 0.9]
  },

  "weights": { "f_per": 0.2, "f_geo": 0.2, "f_ms": 0.6 },

  "scores": {
    "MLP":         0.74,
    "CNN":         0.79,
    "GNN":         0.68,
    "Transformer": 0.91
  },

  "selected": "Transformer",

  "config": {
    "d_model": 32,
    "nhead": 2,
    "ffn_hidden": 512,
    "activation": "gelu",
    "sequence_encoding": "space-time tokens with sinusoidal
    positional encodings"
  }
}
```

## 5.3 CODE AGENT

The Code Agent takes the analysis XML and PINN architecture templates as input and fills modular code blocks (net, loss, data, train, eval, main) to produce a runnable PINN implementation whose residual is symbolically consistent with the governing PDE.

### 5.3.1 SYSTEM PROMPT

**Code Agent — System Prompt**

```
You are the Code Agent in a multi-agent PDE-to-PINN system. Your
    task is to
generate fully runnable PINN code blocks from templates and
    PDE-XML, ensuring
that the PDE residual is symbolically equivalent to the governing
    equation.

You must:
- Generate modular code blocks: [net, loss, data, train, eval,
    main].
- Preserve all function/class names, signatures, and template
    structures.
- Fill all placeholders {...} without deleting or renaming supplied
    templates.
- Implement residual(u, x, t, ...) as a single algebraic expression
    derived
  from the PDE-XML using differentiable operators (du_dx, du_dt,
    ...).
- Ensure residual(...) is symbolically equivalent to the PDE
    (operator level).
- Produce only the completed code blocksno commentary or Markdown
    fences.
- Never output a full monolithic script; only fill the template
    blocks.
- main() must only assemble: generate_data  train  evaluate.
- Set epochs = 10 unless otherwise specified.

Return ONLY the filled block template.
```

### 5.3.2 USER PROMPT

**Code Agent — User Prompt**

```
You are an expert in PINN code generation. Below is the analysis
    XML (with IC/BC)
and the required network implementation.

RULES:
- Generate modular blocks: [net, loss, data, train, eval, main].
- Each block must be fully completed with {} placeholders filled.
- Strictly respect the interface contract implied by the templates:
  * net: define class PINN(nn.Module) ......
  * loss: define a PINN loss that takes model outputs and data
    batches.
  * data: generate_data() returns all tensors needed by train/eval.
  * train: train(model, data, ...) runs the training loop and
    returns the model.
  * eval: evaluate(model, data, ...) returns metrics.
  * main: no new arguments; only wires generate_data, train,
    evaluate.
  Do not change function/class names or argument lists in the
    template.
- No extra explanation, no markdown fences.
- Do NOT generate a monolithic script; only fill the given block
    template.
  Do not redefine main or PINN outside the provided blocks.
```

```
 - All function/class names must be compatible with the given
   network template.
 - For main: do not redefine; only assign hyper-parameters and call
   generate_data, train, evaluate in order.
 - Set epochs = 10.
 - Implement the PDE residual in a function residual(u, x, t, nu,
    ...)
   that is directly derived from the PDE in the XML, using a single
    algebraic
   expression (with +, -, *, /, ** and derivatives). The residual
    must be
   symbolically equivalent to the PDE so that an external checker
    can parse
   residual(...) back into a symbolic form and verify consistency.

# PDE-XML:
{analysis_xml}

#Templates (MUST MATCH):
{net_template},{loss_template},{data_template},
{train_template},{eval_template},{main_template}

# Block template to fill:
{template}
```

### 5.3.3 ASSISTANT OUTPUT

**Code Agent — Assistant Output**

```python
}
import numpy as np
import torch
import torch.nn as nn
# ... (imports, util, PINNsformer definition, etc.)

# ==================== net ====================
class WavePINNsformer(nn.Module):
    """
    net block
    """
    def __init__(self, d_out=1, d_hidden=512, d_model=32, N=1,
    heads=2):
        super().__init__()
        self.model = PINNsformer(
            d_out=d_out,
            d_hidden=d_hidden,
            d_model=d_model,
            N=N,
            heads=heads,
        )

    def forward(self, x, t):

        return self.model(x, t)

def init_weights(m):
    if isinstance(m, nn.Linear):
        torch.nn.init.xavier_uniform_(m.weight)
        if m.bias is not None:
            m.bias.data.fill_(0.01)
```

```python
# ==================== loss ====================
def compute_pde_loss(model, x_res, t_res):
    """
    PDE residual block: u_tt - 4 u_xx = 0
    """
    pred_res = model(x_res, t_res)

    u_x = torch.autograd.grad(
        pred_res, x_res,
        grad_outputs=torch.ones_like(pred_res),
        retain_graph=True, create_graph=True
    )[0]
    u_xx = torch.autograd.grad(
        u_x, x_res,
        grad_outputs=torch.ones_like(pred_res),
        retain_graph=True, create_graph=True
    )[0]

    u_t = torch.autograd.grad(
        pred_res, t_res,
        grad_outputs=torch.ones_like(pred_res),
        retain_graph=True, create_graph=True
    )[0]
    u_tt = torch.autograd.grad(
        u_t, t_res,
        grad_outputs=torch.ones_like(pred_res),
        retain_graph=True, create_graph=True
    )[0]

    # PDE: u_tt - 4 u_xx = 0
    loss_res = torch.mean((u_tt - 4 * u_xx) ** 2)
    return loss_res, pred_res

def compute_bc_loss(model, x_upper, t_upper, x_lower, t_lower):
    """
    Boundary conditions block:
     u(0, t) = u(1, t) = 0
    """
    pred_upper = model(x_upper, t_upper)
    pred_lower = model(x_lower, t_lower)

    loss_bc = torch.mean(pred_upper ** 2) + torch.mean(pred_lower
    ** 2)
    return loss_bc, pred_upper, pred_lower

def compute_ic_loss(model, x_left, t_left, pi):
    """
    Initial conditions block:
      IC1: u(x, 0) = sin(pi x) + 0.5 sin(3 pi x)
      IC2: u_t(x, 0) = 0
    """
    pred_left = model(x_left, t_left)

    # u_t(x, 0)
    u_t_left = torch.autograd.grad(
        pred_left, t_left,
        grad_outputs=torch.ones_like(pred_left),
        retain_graph=True, create_graph=True
    )[0]
```

```
        # t = 0
        u0_pred = pred_left[:, 0]    # [N, 1] -> [N]
        x0 = x_left[:, 0]

        u0_true = torch.sin(pi * x0) + 0.5 * torch.sin(3 * pi * x0)
        loss_ic_1 = torch.mean((u0_pred - u0_true) ** 2)
        loss_ic_2 = torch.mean(u_t_left ** 2)

        loss_ic = loss_ic_1 + loss_ic_2
        return loss_ic, pred_left, u_t_left

    def compute_total_loss(model, batch, pi, loss_track=None):
        """

          L_total = L_res + L_bc + L_ic
        """
        x_res, t_res = batch["x_res"], batch["t_res"]
        x_left, t_left = batch["x_left"], batch["t_left"]
        x_right, t_right = batch["x_right"], batch["t_right"]
        x_upper, t_upper = batch["x_upper"], batch["t_upper"]
        x_lower, t_lower = batch["x_lower"], batch["t_lower"]

        loss_res, _ = compute_pde_loss(model, x_res, t_res)
        loss_bc, _, _ = compute_bc_loss(model, x_upper, t_upper,
        x_lower, t_lower)
        loss_ic, _, _ = compute_ic_loss(model, x_left, t_left, pi)

        total_loss = loss_res + loss_bc + loss_ic

        if loss_track is not None:
            loss_track.append([
                loss_res.item(),
                loss_ic.item(),
                loss_bc.item()
            ])

        return total_loss, loss_res, loss_ic, loss_bc

    # ===================== data / train / eval / main
        =====================

    #   - build_dataset(...)
    #   - train(...)
    #   - u_analytic(...), evaluate_and_plot(...)
    #   - main()

    ...
```

## 5.4 FEEDBACK AGENT

The Feedback Agent executes the code produced by the Code Agent, extracts runtime and numerical signals, and decides whether the pipeline is valid, faulty, or requires refinement, routing structured instructions back to the appropriate agent.

### 5.4.1 SYSTEM PROMPT

**Feedback Agent — System Prompt**

```
You are the FEEDBACK AGENT in a modular PDE-to-PINN system.

Your responsibility is to:
- execute the code file produced by the Code Agent,
- extract runtime signals,
- evaluate the numerical behavior using standardized metrics,
- determine whether the pipeline is VALID, FAULTY, or REQUIRES
    REFINEMENT,
- and route actionable instructions to the appropriate agent.

RUNTIME RESPONSIBILITIES

1. Execute the generated script inside a controlled sandbox.
2. Capture:
   - stdout
   - stderr (full traceback)
   - return code
   - runtime warnings
3. Detect failure conditions:
   - SyntaxError / ImportError
   - shape/type errors
   - divergence: NaN/Inf in losses, exploding gradients
   - runtime timeout
   - missing outputs

METRIC EXTRACTION

If execution succeeds, normalize four metrics into [0,1]:
- m_conv : convergence efficiency (speed of loss reduction)
- m_acc  : accuracy (PDE residual or RelL2)
- m_comp : model complexity (params/FLOPs inversely mapped)
- m_rob  : robustness (gradient stability, smoothness)

Aggregate:
    S(C) = w1*m_conv + w2*m_acc + w3*m_comp + w4*m_rob

ROUTING LOGIC

If status = FAIL:
   - identify the faulty block  {net, loss, data, train, eval,
   main}
    - return an instruction for Code Agent with:
        block_name, error_summary, log_path

If status = SUCCESS but S(C) < threshold:
   - return refinement request (hyperparameter retuning or
     architecture reconsideration) for PINN Agent.

Output MUST be a JSON object.
NO additional text is allowed.
```

### 5.4.2 USER PROMPT

**Feedback Agent — User Prompt**

```
Evaluate the generated PINN code.

Steps:
1) Save the received code into:
      pinn_eval.py

2) Run with:
      subprocess.check_output([...], timeout=120)

3) If execution FAILS:
      - capture stderr (string)
      - save full logs to logs/pinn_eval_*.log
      - analyze traceback to localize the faulty block:
            ["net", "loss", "data", "train", "eval", "main"]
      - return:
        {
          "status": "FAIL",
          "metrics": null,
          "score": null,
          "error_block": "...",
          "error_summary": "...",
          "log_path": "...",
          "action": "call CodeAgent with error_block, log_path,
    error_summary"
        }

4) If execution SUCCEEDS:
      - extract metrics:
        {
          "m_conv": float,
          "m_acc":  float,
          "m_comp": float,
          "m_rob":  float
        }
      - compute S(C)
      - if S(C) >= threshold:
          return {
            "status": "SUCCESS",
            "metrics": {...},
            "score": S,
            "action": "accept"
          }
      - else:
          return {
            "status": "SUCCESS",
            "metrics": {...},
            "score": S,
            "action": "request PINNAgent refinement"
          }

Output MUST be valid JSON.
```

### 5.4.3 ASSISTANT OUTPUT

**Feedback Agent — Assistant Output**

```
{
  "status": "SUCCESS",
  "metrics": {
    "m_conv": 0.87,
    "m_acc": 0.82,
    "m_comp": 0.91,
    "m_rob": 0.88
  },
  "score": 0.87,
  "error_block": null,
  "error_summary": null,
  "log_path": "logs/pinn_eval.log",
  "action": "accept"
}
```

