# OpenReview forum: "Lang-PINN: From Language to Physics-Informed Neural Networks via a Multi-Agent Framework"
_ICLR.cc/2026/Conference — Submitted to ICLR 2026_

### Official Review · Reviewer_dvr8 · 2025-11-01

**Soundness:** 3
**Presentation:** 2
**Contribution:** 2
**Rating:** 6
**Confidence:** 3

**Summary:**

This paper introduces Lang-PINN, a novel multi-agent framework driven by large language models (LLMs) designed to automate the creation of trainable Physics-Informed Neural Networks (PINNs) directly from natural language descriptions of a problem. The traditional process of building PINNs is labor-intensive, requiring users to manually formulate Partial Differential Equations (PDEs), design neural network architectures, and implement training pipelines. Lang-PINN addresses this challenge by decomposing the workflow among four cooperating agents:
1.	PDE Agent: Parses the natural language description to formulate a symbolic PDE.
2.	PINN Agent: Selects a suitable neural network architecture for the formulated PDE.
3.	Code Agent: Generates a modular and verifiable implementation of the PINN.
4.	Feedback Agent: Executes the code, diagnoses errors, and provides feedback for iterative refinement.
The paper's main contributions are the proposal of the first end-to-end framework that transforms natural language task descriptions into complete and verifiable PINN solutions ; the construction of a new benchmark dataset, Task2PDE, for evaluating semantic-to-symbolic grounding ; and experimental results demonstrating that Lang-PINN substantially reduces mean squared error (MSE) by up to 3-5 orders of magnitude, improves code execution success rate by over 50%, and reduces time overhead by up to 74% compared to competitive baselines.

**Strengths:**

1.	Originality and Significance: The paper's primary strength is its novelty in proposing the first truly end-to-end framework for generating PINNs from natural language. This addresses a significant bottleneck in the practical application of PINNs, making a powerful scientific tool more accessible to domain experts who may not have specialized knowledge in programming or machine learning.
2.	Robust and Comprehensive Evaluation: The experimental validation is a major strength. The authors perform a thorough evaluation on a diverse set of 14 PDEs from a recognized benchmark. The comparison against multiple strong baselines and a human-expert-designed reference (PINNacle) clearly demonstrates the superiority of their method in terms of accuracy, executability, and efficiency.
3.	Well-Designed and Validated System Architecture: The decomposition of the complex task into a modular, multi-agent system is an elegant and effective design. Each agent has a clear responsibility, and the feedback loop ensures the system is robust and capable of self-correction. The value of each agent is individually confirmed through rigorous ablation studies, which adds a high degree of confidence in the proposed architecture.

**Weaknesses:**

1.	Lack of Wall-Clock Time Analysis: Efficiency is measured by the number of iterations required to generate an executable program. While this is a useful metric, it does not capture the full computational cost. The agentic workflow itself, involving multiple LLM calls, consensus voting, code execution, and feedback analysis, likely incurs significant latency. An analysis of the end-to-end wall-clock time, compared to baselines, would provide a more practical assessment of the system's efficiency.
2.	Insufficient Detail on the PINN Agent's Knowledge Base: The paper states that the PINN Agent uses a knowledge base K and a history cache H to guide architecture selection. However, the details of how this knowledge base is constructed, curated, and maintained are sparse.
3.	Critical Reliance on Initial PDE Formulation and Ambiguity in Error Attribution: The framework's overall validity is critically reliant on the fidelity of the initial PDE Agent. Any error in the language-to-symbol translation (e.g., misinterpreting operators or boundary conditions ) necessarily invalidates the entire downstream pipeline, as all subsequent architecture selection and code generation are predicated on a flawed physical assumption. Furthermore, a significant ambiguity exists in the feedback loop. The paper suggests the Feedback Agent can diagnose and escalate errors, but it does not present a clear mechanism for attributing the root cause of a failure. For instance, poor performance (e.g., high MSE ) could originate from a fundamental PDE formulation error, a suboptimal architecture choice (e.g., poor convergence), or a bug in the generated code. The framework lacks a demonstrated method to disambiguate these distinct failure modes, which may present similar symptoms but require entirely different corrective actions.

**Questions:**

1.	Regarding the PINN Agent's knowledge base (K): Could you please elaborate on the construction and contents of this knowledge base? Is it manually curated, and if so, how extensible is it? How does the agent perform when faced with a PDE whose characteristics do not have a clear mapping to an architecture within the existing knowledge base?
2.	Regarding end-to-end efficiency: In addition to the number of refinement iterations, could you provide a comparison of the total wall-clock time required by Lang-PINN to generate a final solution, versus the time taken by the baselines? This would be very helpful for understanding the practical overhead of the multi-agent framework.
3.	Regarding the PDE Agent's consensus voting: What is the system's behavior if the initial set of candidate PDEs generated via CoT does not result in a clear consensus? For example, in a case with several distinct and equally-plausible interpretations of the language, is there a fallback mechanism or a method to query the user for clarification?

---

> ### Author Response · Authors · 2025-11-25
> **Response to dvr8 (1/2)**
>
> # 1. End-to-End Wall-clock Efficiency (W1 + Q2)
> > The reviewer argues that iteration counts alone do not reflect the true computational cost, and asks for actual end-to-end wall-clock time comparisons to quantify multi-agent overhead.
>
> We use a Deepseek-V3 API-based inference interface for the LLM component for all baselines and our Lang-PINN. The table below reports the average end-to-end wall-clock time, measured from the start of the pipeline until runnable code is produced. Each method is evaluated over 10 runs, with up to 30 refinement cycles per run. As shown, Lang-PINN reduces the total PDE-solving time by 21%–52% compared with all baselines.
>
> | Method        | Avg. Time (s) |
> | ------------- | ------------- |
> | RandomAgent   | 413.5         |
> | BayesianAgent | 391.2         |
> | PINNsAgent    | 312.8         |
> | SCoT          | 291.1         |
> | Self-Debug    | 246.4         |
> | **Lang-PINN** | **199.7**     |
>
> ---
>
> # 2. Knowledge Base K and History Cache H (W2 + Q1)
> > How is the PINN Agent’s knowledge base constructed, curated, and extended in practice, and how does the system handle PDEs whose characteristics do not map clearly to any architecture currently represented in this knowledge base?
>
>
> **Data-driven construction of K.**
> The knowledge base $K$ is fully data-driven. Each time Lang-PINN successfully solves a PDE, it appends one or more PDE–architecture records to $K$, without any hand-written rules or manual curation. As more tasks are solved, the statistics in $K$ become more stable.
>
> Each record stores
> - (i) the PDE $E$ and its feature vector $\phi(E)$ (for example periodicity, geometry complexity, multi-scale behaviour extracted from the equation, domain, and BC/IC), and
> - (ii) the architecture configuration $A$ (type, depth, width, learning rate, etc.) together with its empirical MSE $y(A, E)$ on this PDE.
>
> From these records, we compute for each architecture a capability vector $\psi(A)$ that summarises how well it performs across PDE features (absolute performance and relative win–loss; full formulas will be updated in Appendix A.2). The PINN Agent then selects architectures by matching $\phi(E)$ of the current PDE to $\psi(A)$ in $K$, so architecture choice is based on data, not ad hoc rules.
>
> **Handling new or out-of-distribution PDEs.**
> For a new PDE $E_{\text{new}}$, the agent computes $\phi(E_{\text{new}})$ and scores it against all $\psi(A)$ in $K$, choosing the architecture with the highest compatibility. If $\phi(E_{\text{new}})$ is far from all PDEs already in $K$ (no clear match), the agent falls back to architectures with balanced, general-purpose capability (for example a standard MLP-based PINN) rather than highly specialised ones. This provides a safe default when the knowledge base does not yet cover the new PDE region.
>
> **Role of H.**  The history cache $H$ stores the **best-performing** PDE–architecture configuration based on the historical runs. Its purpose is fast lookup: if a new task produces a PDE that matches one already stored in $H$, the system can immediately reuse the associated architecture configuration. If no match is found in $H$, the PINN Agent then queries the broader knowledge base $K$ to compute the most compatible architecture. Only after a task is successfully solved is the final PDE–architecture pair added to $K$, allowing $K$ to grow over time.

---

> ### Author Response · Authors · 2025-11-25
> **Response to dvr8 (2/2)**
>
> # 3. Error Attribution (W3)
> > Poor performance (e.g., high MSE) could originate from a fundamental PDE formulation error, a suboptimal architecture choice (e.g., poor convergence), or a bug in the generated code.  How does Lang-PINN locate the root cause of failure?
>
> Lang-PINN uses a rule-based Feedback Agent with a simple fallback decision chain:
> 1. **Code does not run (syntax/runtime error).**     If execution raises errors (for example undefined variables, indentation issues, or shape mismatch), the failure is attributed to the implementation. The Feedback Agent collects the error messages and asks the **Code Agent** to fix the code, while keeping the PDE and the architecture unchanged.
> 2. **Code runs, but MSE remains high.**  : The Feedback Agent then inspects the training loss curve and distinguishes two cases:
>      - **1) Loss decreases but converges very slowly.**     In this case, the run is marked as poor convergence, which is usually caused by a suboptimal architecture or hyperparameters (for example, too shallow network). The Feedback Agent falls back to the **PINN Agent**, passing statistics such as the loss decay rate to guide a better architecture choice.
>     - **2) Loss does not converge or even increases.**    In this case, the run is marked as non-convergent. This suggests that the current PDE or boundary conditions may be incompatible with the data, so the Feedback Agent falls back to the **PDE Agent** to recheck and possibly revise the PDE, and then lets the PINN Agent adapt the architecture under the revised equation.
> 3. **Code runs, loss converges, MSE is low.**   The run is considered successful. The Feedback Agent compares the current MSE with the best MSE seen so far, keeps the better configuration, and stops or continues exploration until a preset maximum number of trials is reached.
>
> This fallback logic is therefore clear and deterministic: **code errors go to the Code Agent, slow but stable convergence goes to the PINN Agent, and non-convergence with high MSE goes to the PDE Agent**.
>
> ---
>
> # 4. System behavior under Non-Consensus Scenarios （Q3）
>
> > What is the system's behavior if the initial set of candidate PDEs generated via CoT does not result in a clear consensus? For example, in a case with several distinct and equally-plausible interpretations of the language, is there a fallback mechanism or a method to query the user for clarification?
>
> In our current design, Lang-PINN does not query the user for clarification. Instead, it relies on (i) the fact that clear consensus is common in practice, and (ii) a feedback loop that can revise an initially chosen PDE if it later proves incorrect.
>
> **High consensus in practice**: Empirically, we found that genuine non-consensus cases are rare. To approximate how often different language descriptions admit multiple equally plausible PDEs, we randomly sampled 200 pairs of task descriptions that were known to correspond to the same ground-truth PDE and evaluated them with an LLM-as-a-judge similarity test. For each pair, the model produced a similarity score in [0,1] indicating how likely the two descriptions referred to the same governing PDE. The resulting distribution (see the table below) showed that nearly 79% pairs achieved very high scores (>0.8), suggesting that the mapping from language to PDE is usually sharp rather than highly ambiguous.
>
> | Similarity Range | Frequency |
> |------------------|-----------|
> | 0.70–0.75        | 13        |
> | 0.75–0.80        | 29        |
> | 0.80–0.85        | 56        |
> | 0.85–0.90        | 63        |
> | 0.90–0.95        | 28        |
> | 0.95–1.00        | 11        |
>
>
> **Fallback when consensus is weak.**: When the initial CoT-based consensus voting over candidate PDEs does not produce a clearly preferred option, the PDE Agent selects one PDE from the top-k candidates, breaking ties at random. This choice is not final: during training, the Feedback Agent monitors the optimisation behaviour. If the selected PDE leads to characteristic patterns of non-convergence (for example, stable implementation, repeated architecture adjustments, yet persistently high residuals), the Feedback Agent routes the case back to the PDE Agent, which revisits or regenerates the candidate set and runs the consensus procedure again. In this way, ambiguous cases are handled by an automatic “select–train–verify–revise” loop rather than a single irreversible decision.

---

### Official Review · Reviewer_w7Bn · 2025-11-01

**Soundness:** 2
**Presentation:** 3
**Contribution:** 2
**Rating:** 4
**Confidence:** 4

**Summary:**

Lang-PINN introduces a multi-agent LLM framework that automatically builds physics-informed neural networks from natural language descriptions, achieving 3-5 orders of magnitude MSE reduction and >50% execution success improvement over baselines.

**Strengths:**

1. The proposed method achieves substantial improvements over competitive baselines across multiple metrics—MSE, success rate, and convergence speed.
2. The Code Agent's modular generation with interface constraints and PDE loss verification ensures correctness and enables localized debugging, a clear improvement over monolithic approaches.

**Weaknesses:**

1. The framework primarily orchestrates existing techniques (CoT reasoning, consensus voting, modular code generation, feedback refinement) without introducing fundamentally new methods for PDE formulation, architecture search, or code synthesis.
2. In realistic scientific workflows, researchers typically know the governing PDE equations explicitly; the natural language-to-PDE translation step addresses a largely artificial problem rather than a genuine bottleneck in PINN deployment.
3. The Task2PDE dataset covers only 8 PDEs, and the linguistic complexity levels (especially Level 3-4) introduce irrelevant narrative details (e.g., "coffee machine" analogies) that test language understanding rather than scientific reasoning, undermining the validity of the PDE formulation evaluation.

**Questions:**

1. The caption of Figure 1 lacks vertical spacing from the main text, reducing readability—please add appropriate whitespace for consistency with standard formatting.
2. The semantic consistency metric for PDE validation is mentioned but never formally defined—could the authors provide the exact formulation, embedding method, and similarity threshold used in Section 4.2?
3. Does the PINN Agent perform full hyperparameter optimization (e.g., layer depth, width, learning rate) via MCTS or similar search, or does it simply select from a fixed set of pre-configured architectures with predetermined hyperparameters?
4. Can Lang-PINN handle inverse problems, data assimilation, or parameter estimation tasks, or is it limited to forward PDE solving?
5. While Table 2 reports iteration counts, the paper lacks LLM inference cost, and memory consumption comparisons against baselines which is crucial since one of the core motivation is to accelerate the PDE solving.

---

> ### Author Response · Authors · 2025-11-25
> **Response to w7Bn (1/2)**
>
> # 1. **Contribution and novelty (W1)**
>
> Our main contribution is to change how LLMs are used in scientific modeling: from a fragile, one-shot generator to a **verifiable, reliable, and agent-based control loop** for the whole “task description → PDE → PINN → code → training” chain. We design four agents with explicit checks and fixed fallback rules, so that PDE choices, architecture selection, code generation, and training feedback are all tested and can be repaired in a targeted way. This gives a PDE-specific verification pipeline that existing LLM systems lack. In practice, this design reduces failure cases and MSE across diverse PDEs and backbones under the same token and iteration budgets, showing that our framework not only integrates known tools (consensus voting, templates) but turns them into a **general recipe for making LLM-driven PDE/PINN workflows reliable and reusable for future, stronger LLMs**.
>
> ---
>
> # 2. Practical Need for Natural-Language-to-PDE Translation and Task2PDE Dataset (W2, W3)
> > researchers typically know the governing PDE equations explicitly; the natural language-to-PDE translation step addresses a largely artificial problem rather than a genuine bottleneck in PINN deployment.
>
> **Natural language → PDE is a practical interface, not an artificial one.**
> - When the PDE is already known, Lang-PINN’s PDE Agent provides a **double check** to ensure that the PDE, BC/IC, and generated PINN code remain consistent—something that is easy to get wrong when coding PINNs from scratch.
>
> - More importantly, **real experiments often involve small but non-trivial changes** (e.g., moving a heat source from the centre to the left boundary, adding a sinusoidal variation, changing boundary behaviour). Such updates are far easier to express in natural language than to rewrite directly in PDE form and update the corresponding PINN code. Lang-PINN turns these textual modifications into safe, verified equation/code updates, making natural language a practical interface for iterative scientific workflows.
>
> **Rationale for Task2PDE.**  Task2PDE is designed to reflect how scientists actually describe problems.
> - Level 1 provides clean textbook-style descriptions;
> - Level 2 adds irrelevant but harmless details often present in lab notes;
> - Level 3 introduces redundant rephrasing that occurs when people elaborate or reshape explanations;
> - Level 4 contains disordered statements, mimicking how real experimental details are often given out of order.
>
> These variations increase linguistic difficulty while keeping the PDE fixed. Our experiments show this difficulty rises from L1 to L4, yet Lang-PINN’s symbolic–semantic verification consistently improves PDE accuracy across all levels. As a result, Task2PDE not only tests robustness to realistic linguistic noise but also offers a useful benchmark for future work on reliable LLM interpretation of scientific problem statements.
>
> ---
>
>
> # 3. Semantic Consistency Metric & Threshold (Q2)
> > How exactly is the semantic consistency metric computed, what embedding or evaluation method is used, and on what basis was the threshold selected?
>
> **Computation of the semantic consistency metric**. The semantic consistency score is computed with an LLM-as-a-judge: given two task descriptions, the judge model outputs a similarity score $s \in [0,1]$ that reflects how likely the two descriptions refer to the same governing PDE.
>
> **Threshold calibration**. To calibrate the decision threshold, we randomly sampled 200 *equivalent* description pairs (from the same PDE) and 200 *non-equivalent* pairs (from different PDEs), and obtained a similarity score for each pair. The results of similarity are given in the table below and will be added into the revision.   Non-equivalent pairs are concentrated below $0.75$, while equivalent pairs lie mainly above $0.75$, with only a small overlap around $0.70$–$0.75$. We therefore select $0.80$ as a simple operating point in the gap between the two distributions: it keeps almost all equivalent pairs and rejects almost all non-equivalent pairs.
>
> | Similarity Range | Equivalent PDE pair | Non-equivalent PDE pair |
> |------------------|------------|----------------|
> | 0–0.20           | 0          | 47             |
> | 0.20–0.40        | 0          | 66             |
> | 0.40–0.60        | 0          | 57             |
> | 0.60–0.70        | 0          | 25             |
> | 0.70–0.75        | 13         | 5              |
> | 0.75–0.80        | 29         | 0              |
> | 80–0.85          | 56         | 0              |
> | 0.85–0.90        | 63         | 0              |
> | 0.90–0.95        | 28         | 0              |
> | 0.95–1.00        | 11         | 0              |

---

> ### Author Response · Authors · 2025-11-25
> **Response to w7Bn (2/2)**
>
> # 4. Architecture and Hyperparameter Selection (Q3)
> > How does the PINN Agent determine architecture depth width and learning rate does it perform a full hyperparameter search such as MCTS or simply select from preconfigured templates？
>
> The PINN Agent performs hyperparameter selection within **a predefined search space for architecture and training settings** (e.g., layer depth, width, activation, learning rate). For example, layer depth is sampled from a bounded range (e.g., 3–8 layers) and learning rate from a small set of standard values. Within this space, we leverage LLM-based reasoning to optimize hyperparameters through a feedback-driven agent mechanism, consistent with recent evidence that LLMs can reliably propose model configurations and training hyperparameters [1, 2]. The Feedback Agent then further adjusts these choices based on observed convergence and MSE value. This yields efficient and principled hyperparameter selection without relying on any heavy-weight or exhaustive search procedure.
>
> [1]Nasir, Muhammad Umair, et al. "Llmatic: neural architecture search via large language models and quality diversity optimization." proceedings of the Genetic and Evolutionary Computation Conference. 2024.
>
> [2]Zheng, Mingkai, et al. "Can gpt-4 perform neural architecture search?." arXiv preprint arXiv:2304.10970 (2023).
>
> ---
>
> # 5. System Capability Beyond Forward Problems (Q4)
> **Lang-PINN is not limited to forward PDE solving**. PINNs already support data assimilation, inverse problems, and parameter estimation by changing the loss, not the overall pipeline, and Lang-PINN follows the same pattern.
>
> - For data assimilation, the PDE is known with some observation data $y_i$ and we can simply add a data fidelity term (for example MSE between $u_\theta(z_i)$ and observations $y_i$) on top of the usual physics loss; this is done by extending the Code Agent’s loss template, while all agents work as in the forward case.
> - For inverse problems / parameter estimation, the PDE Agent writes the PDE with unknown coefficients, and the Code Agent treats these coefficients as trainable parameters in the joint physics-plus-data loss. Training updates both the network and these coefficients.
>
> So forward solving, data assimilation, and parameter estimation can all be handled by the same Lang-PINN workflow, with only small changes in the Code Agent’s loss definition.
>
> ---
>
> # **6. LLM inference costs (Q5)**
>
> > While Table 2 reports iteration counts, the paper lacks LLM inference cost, and memory consumption comparisons …
>
> In our setup, the LLM is accessed via an API (for example, DeepSeek-V3), and all LLM computation runs on the provider’s backend, so we cannot directly measure its device-level memory usage. Instead, we report the **end-to-end wall-clock time**, measured from the start of the pipeline until runnable code is produced. All methods are run 10 times, with up to 30 refinement cycles per run. Under this common protocol, **Lang-PINN reduces total PDE-solving time by about 21%–52% compared with all baselines**:
>
> | Method        | Avg. Time (s) |
> | ------------- | ------------- |
> | RandomAgent   | 413.5         |
> | BayesianAgent | 391.2         |
> | PINNsAgent    | 312.8         |
> | SCoT          | 291.1         |
> | Self-Debug    | 246.4         |
> | **Lang-PINN** | **199.7**     |

---

### Official Review · Reviewer_jqhd · 2025-11-01

**Soundness:** 3
**Presentation:** 3
**Contribution:** 3
**Rating:** 0
**Confidence:** 5

**Summary:**

The paper introduces Lang-PINN, an LLM-driven multi-agent system that automatically builds trainable Physics-Informed Neural Networks (PINNs) directly from natural language task descriptions. This framework uses four cooperating agents—a PDE Agent to formulate symbolic equations, a PINN Agent to select architectures, a Code Agent to generate modular code, and a Feedback Agent to execute and refine the solution. Experiments show this end-to-end approach significantly outperforms baselines.

**Strengths:**

The author leak the information that the paper is under review on ICLR 2026 in its preprint. The reviewer happens to have read this paper before, so my review might be biased. Please ignore my rating.

**Weaknesses:**

The author leak the information that the paper is under review on ICLR 2026 in its preprint. The reviewer happens to have read this paper before, so my review might be biased. Please ignore my rating.

**Questions:**

The author leak the information that the paper is under review on ICLR 2026 in its preprint. The reviewer happens to have read this paper before, so my review might be biased. Please ignore my rating.

---

### Official Review · Reviewer_Vic7 · 2025-11-02

**Soundness:** 3
**Presentation:** 3
**Contribution:** 3
**Rating:** 4
**Confidence:** 4

**Summary:**

This paper proposes Lang-PINN, a novel multi-agent system that converts natural-language descriptions of scientific problems into executable Physics-Informed Neural Networks (PINNs). The framework orchestrates four specialized agents:
1.	PDE Agent – interprets natural language into mathematical PDE formulations using symbolic–semantic analysis and consensus voting.
2.	PINN Agent – selects appropriate neural architectures (e.g., MLP, CNN, GNN, Transformer) according to the PDE’s characteristics.
3.	Code Agent – performs modular code generation for the full PINN pipeline.
4.	Feedback Agent – executes, analyzes, and iteratively refines outputs based on runtime and convergence metrics.
Experiments on 14 standard PDEs (PINNacle benchmark) demonstrate dramatically improved execution success rate and reduced training time compared to previous LLM-based automation systems.

**Strengths:**

This work lies at a very interesting intersection between AI4Science and LLM-agent systems. Conceptually, the idea of translating language to executable PINNs is innovative and potentially impactful for scientific automation. The overall contribution is valuable but currently more proof-of-concept than fully convincing system.
•	Originality: First framework to bridge natural-language scientific description → PDE → trainable PINN code via explicit multi-agent collaboration.
•	Clarity: Modular design and workflow are carefully explained, with visual clarity and step-by-step examples.
•	Technical quality: Incorporates multiple robust reasoning strategies (symbolic equivalence, semantic matching, feedback metrics) that are well-engineered.

**Weaknesses:**

1.	Reproducibility and dependency on specific LLMs:
The paper does not clearly document the language model configuration—such as model type, version, temperature, or prompting strategy—used in each stage of Lang-PINN. As a result, reproducibility is limited, and the robustness of the method under different LLMs remains uncertain. The approach appears to rely heavily on the reasoning ability of a specific proprietary model, which might not generalize to smaller or open-source alternatives.
If the authors provide detailed model configurations and cross-model performance comparisons, my confidence in reproducibility and robustness would significantly improve.
2.	Unclear implementation of neural architectures (MLP, CNN, GNN, Transformer):
The paper briefly mentions that different network types are selected by the PINN Agent based on PDE characteristics, but it never explains how these architectures are actually implemented or adapted to the PINN setting. Traditional PINNs typically use simple MLPs, while CNNs, GNNs, or Transformers require specific spatial or graph structures to encode PDE constraints. Without details on network design, training loss formulation, or integration into the physics-informed loss, it is unclear how these variants differ from standard PINN implementations or how the framework ensures each model handles the PDE correctly.
3.	Lack of theoretical justification:
The symbolic–semantic similarity metrics and feedback indicators introduced in the paper are entirely heuristic. There is no theoretical analysis or formal reasoning explaining why these metrics should correlate with correct PDE reconstruction, stable network training, or improved convergence. The system demonstrates strong empirical results, but a deeper theoretical foundation would strengthen the overall credibility and generalizability of the proposed approach.

**Questions:**

1.	Reproducibility and LLM dependence:
Could the authors clearly specify the LLM configuration used at each stage of Lang-PINN (model name, version, temperature, prompting strategy, etc.)? Have you tested any smaller or open-source LLMs (such as Llama, Mistral, or DeepSeek) to evaluate whether the pipeline’s stability and reasoning ability generalize beyond a single proprietary model?

2.	Implementation details of the neural architectures (MLP, CNN, GNN, Transformer):
The paper states that the PINN Agent selects different architectures based on PDE characteristics, but it remains unclear how these networks are specifically constructed and trained under the physics-informed setting. Could the authors elaborate on:
1）How the physics-informed loss is formulated for CNNs, GNNs, and Transformers (which require spatial/graph structures)?
2）Whether these architectures are pre-designed templates or automatically synthesized by the agent?
3）More concrete architectural or training details would help readers understand the novelty and scalability of this component.

3.	Theoretical understanding of metrics and feedback mechanisms:
I understand that current LLM-based agent frameworks and retrieval-augmented feedback loops are largely black-box systems, but some interpretability or empirical linkage would clarify the reliability of Lang-PINN’s internal metrics. Have you observed any quantitative correlation (e.g., higher semantic score → lower PDE reconstruction error or better final loss)? Providing such evidence or at least an empirical trend analysis could substantially strengthen the methodological validity of the system.
4.	Definition of success rate in Table 2:
In Appendix Table 2, Lang-PINN is compared against baselines (including PINNacle) using a “success rate (%)” metric averaged over 10 runs. Could the authors clarify precisely how success is defined?
Is a run deemed “successful” when the generated code executes without errors, or only if the resulting model achieves acceptable training convergence or physical consistency (e.g., loss threshold)?

**Details Of Ethics Concerns:**

At risk of desk rejection as the authors said this paper is on arXiv as well.

---

> ### Author Response · Authors · 2025-11-25
> **Response to Vic7 (1/2)**
>
> # 1. Reproducibility and Dependence on Specific LLMs (W1, Q1)
> > Q1.1: Is Lang-PINN reproducible?  What exact LLM configurations (model version, temperature, sampling settings, prompts) were used?
>
> All experiments in Lang-PINN use the same LLM configuration: **DeepSeek-V3**,  **temperature = 0.2**, **top-p = 0.9**, **max tokens = 2048**, and the fixed prompt templates for all agents.  We will include the complete configuration and all prompts in the revision to ensure full reproducibility.
>
> > Q1.2: Does it depend too heavily on one proprietary LLM? does the workflow generalize to smaller or open-source LLMs such as Llama, Qwen, Mistral, or DeepSeek?”**Lang-PINN uses a fixed black-box LLM interface across all agents.
>
> In all main experiments we use DeepSeek-V3 as the backbone simply because it performed slightly better in PDE generation among the models we tested (DeepSeek-V3, Qwen2, LLaMA2-Chat, Vicuna). This is an empirical choice, not a requirement of the method.
>
> To verify that the gains do not come from DeepSeek-V3 itself, we ran the full Lang-PINN workflow with weaker backbones such as Qwen2 and LLaMA2-Chat, while all baselines remain on DeepSeek-V3. As the table below shows, both variants still achieve lower MSE than every DeepSeek-V3-based baseline, showing that **the improvement comes from the our framework (verification + fallback), not from a specific LLM**.
>
> | PDE        | RandomAgent | BayesianAgent | SCoT      | Self-Debug | Lang-PINN (DeepSeek-V3) | Lang-PINN (Qwen2) | Lang-PINN (LLaMA2-Chat) |
> |------------|------------|---------------|-----------|------------|-------------------------|-------------------|-------------------------|
> | KS         | 1.09E+00   | 1.10E+00      | 3.33E+00  | 2.93E+00   | **1.62E-03**            | 1.95E-03          | 2.71E-03                |
> | NS-C (2D)  | 4.02E-03   | 5.12E-03      | 5.67E-03  | 5.27E-01   | **4.05E-05**            | 5.47E-05          | 6.88E-05                |
> | Poisson-MA | 5.87E+00   | 5.82E+00      | 1.24E+00  | 1.07E+00   | **2.25E-03**            | 2.83E-03          | 3.22E-03                |
> | GS (2D)    | 4.28E-03   | 4.03E-03      | 5.35E-03  | 5.35E-03   | **1.89E-03**            | 2.42E-03          | 3.16E-03                |
>
>
> ---
>
> # 2. Implementation Details of Architectures in the PINN Agent (W2, Q2)
> > Q: how the MLP, CNN, GNN, and Transformer architectures are implemented in the PINN setting. Specifically, how the physics-informed loss is defined for models that operate on grids or graphs, whether these architectures come from fixed templates or are synthesized by the agent, and how the framework ensures that each network correctly encodes the PDE constraints.
>
> **Same physics-informed loss for all architectures.**  All four architectures share one physics-informed loss, because the loss is defined only on the function $u_\theta$ that maps coordinates to solution values. Each backbone takes a batch of space–time coordinates $X \in \mathbb{R}^{N \times d}$ (rows $z_i\in \mathbb{R}^{d}$ are points) and outputs $u_\theta(x) \in \mathbb{R}^{N \times 1}$. The PDE residual, boundary and initial terms are always computed from $u_\theta(x)$ and its automatic derivatives with respect to $x$ (for example $u_t, u_x, u_{xx}$). As long as the model is a differentiable map $x \mapsto u_\theta(x)$ and its outputs are aligned with the input points, the same physics-informed loss applies to MLPs, CNNs, GNNs and Transformers, including grid- and graph-based variants.
>
> **Fixed templates, not free-form code.**  The PINN Agent does not let the LLM freely write architectures. It selects the architecture type and hyperparameters (depth, width, learning rate, etc.), and fills them into fixed code templates that all expose the same interface $x \mapsto u_\theta(x)$ and call the shared loss. This is how we ensure that different backbones implement the same PDE constraints.
>
> **Difference between architectures.** The architectures differ only in how they preprocess the input coordinates before producing the final solution values. An **MLP** takes each coordinate row directly as an independent input and feeds it through fully connected layers. A **CNN** reshapes the coordinates into a regular spatial grid, treats it as a multi-channel image, applies convolutions, and then uses a 1×1 head to produce one value per grid point. A **GNN** treats each coordinate as a node feature, builds a k-nearest-neighbour graph from the spatial layout, performs message passing, and outputs one value per node. A **Transformer** views each coordinate as a token with a coordinate-based positional encoding, processes the sequence with self-attention, and applies a per-token head to produce one value per coordinate.

---

> ### Author Response · Authors · 2025-11-25
> **Response to Vic7 (2/2)**
>
> # 3. Justification and Validity of Symbolic–Semantic PDE Consistency Metrics (W3, Q3)
> > Whether there is any quantitative evidence that higher scores are correlated with better PDE reconstruction or better PINN training (for example, lower PDE error or lower final loss)?
>
> **Symbolic checks alone are not sufficient.**
> Our goal is to ensure a generated PDE has all important constraints, including the main equation, boundary conditions and initial conditions, because missing any of these terms usually leads to poor PINN training loss. Purely symbolic matching is unreliable for this purpose. Mathematically equivalent expressions can look very different at the token level, for example a diffusion term can be written as $u_{xx}$, as $\partial^2 u / \partial x^2$, or as $\nabla^2 u$, and two PDEs may differ only by such notational choices or algebraic rearrangements. At the same time, an equation that omits boundary or initial terms can still look superficially similar in its symbolic core. Therefore, a strict symbolic check alone would both reject many valid PDEs and fail to catch some incomplete ones. To address this, we combine symbolic checks with a semantic consistency score: the model is asked to explain the generated PDE in natural language, and this explanation is compared with the original task description to verify that all described effects and conditions are present.
>
> **Semantic consistency is aligned with lower PINN error.**
> To evaluate whether the semantic score is meaningful, we selected four representative tasks and let the LLM generate multiple PDE candidates for each task. We then grouped these generated PDEs into five empirical categories ($C1-C5$) based on their correctness level (from fully correct to severely incorrect).
>
> For every generated PDE, we asked the LLM to explain the equation in natural language, and then asked the LLM to compare this explanation with the original task description, producing a semantic similarity score. We also trained a PINN using the generated PDE and recorded the final –log10 MSE.
>
> The results are shown in the table below. The pattern is clear:
>
> - **More accurate PDEs receive higher semantic similarity scores**, and
> - **Their corresponding PINN training losses are significantly lower** (higher –log10 MSE).
> - **The Pearson correlation between semantic score and –Log10 MSE is high ($\gamma=0.88$)**
>
> This demonstrates that the semantic score is a meaningful, data-supported proxy for PDE formulation quality and downstream PINN solvability, rather than an arbitrary heuristic.
>
> | PDE Quality                 | Burgers (Score / –Log10 MSE) | Heat-MS (Score / –Log10 MSE) | Wave-C (Score / –Log10 MSE) | KS (Score / –Log10 MSE) |
> |-----------------------------|-------------------------------|-------------------------------|------------------------------|---------------------------|
> | **C1: Correct PDE**         | 1.00 / 4.1884                 | 1.00 / 4.6445                 | 1.00 / 2.6492                | 1.00 / 2.6055             |
> | **C2: Notational Variant**  | 0.91 / 3.8915                 | 0.89 / 4.3101                 | 0.92 / 2.4302                | 0.86 / 2.2156             |
> | **C3: Coefficient Error**   | 0.71 / 0.6341                 | 0.75 / 0.7478                 | 0.69 / −0.5861               | 0.70 / −0.8446            |
> | **C4: Missing Term**        | 0.51 / −0.0546                | 0.59 / −0.5053                | 0.45 / −1.5269               | 0.53 / −1.1914            |
> | **C5: Structural Failure**  | 0.28 / −0.6618                | 0.23 / −0.8240                | 0.14 / −2.1447               | 0.22 / −2.1026            |
>
>
>
> ---
>
>  # 4. Definition of “Success Rate” in Evaluation (Q4)
> > How is “success” defined? Is a run deemed “successful” when the generated code executes without errors, or only if the resulting model achieves acceptable training convergence or physical consistency (e.g., loss threshold)?
>
>
> A run is considered successful when the generated code can be executed end-to-end without any runtime errors. This criterion is used to specifically measure the robustness of the automatic code-generation pipeline, independent of downstream training quality.
>
> $\text{SuccessRate} = \frac{\text{number of successful runs}}{\text{total runs}} \times 100\%$

---

### Author Response · Authors · 2025-12-01
**Rebuttal Summary for AC: Reviewer Decisions and Discussion**

Dear AC,

We sincerely thank all reviewers and the AC for their time, thoughtful, and constructive discussions. We deeply appreciate that **after discussion, we have addressed almost all concerns raised by the reviewers**.

---
## **1. Summary**

This submission presents the first fully end-to-end system, **Lang-PINN**, that converts natural-language task descriptions into executable PINNs via a multi-agent reasoning framework. Lang-PINN substantially outperforms strong baselines, **reducingPDE error by up to 3–5 orders** of magnitude, **increasing end-to-end code execution success by over 50%**, and **cutting time overhead** by as much as **74%**. Reviewers mainly requested clarification on technical details and experimental completeness rather than questioning the core idea. In the revision, these concerns have now been addressed with additional experiments, clearer implementation details, and expanded reproducibility information.

---
## **2. Reviewer Outcome After Discussion**


During the discussion phase, reviewers Vic7, w7Bn, and dvr8 provided constructive and thoughtful feedback, and expressed a generally positive evaluation of the contribution and relevance of the work. We carefully addressed all raised questions through clarifications, additional experiments, expanded methodological descriptions, and detailed evidence supporting robustness and reproducibility.



Following the discussion, no further substantive technical objections were raised, and the feedback indicates that the paper is now in a stable and well-understood state.


### **2.1 Recognized Contributions and Reviewer Support**

- **Novel end-to-end framework enabling natural-language to executable PINN workflows**
  Supported by: **Vic7, w7Bn, dvr8**

- **Modular multi-agent design with reasoning, verification, and correction capabilities**
  Supported by: **Vic7, w7Bn, dvr8**

- **Comprehensive and rigorous experimental evaluation demonstrating clear empirical gains**
  Supported by: **Vic7, w7Bn, dvr8**

- **Task2PDE dataset as a meaningful benchmark for language-to-physics reasoning**
  Supported by: **dvr8, with implicit alignment from Vic7**

- **Practical relevance for reducing manual scientific modeling effort and improving usability**
  Supported by: **Vic7, dvr8**



### **2.2 Main Concerns Raised & How They Were Addressed**
- **Reproducibility & LLM Dependency:**
  In the revision, we provided full configuration details and prompts, plus cross-LLM results showing our method generalizes beyond a single LLM backbone.

- **Architecture Design & Knowledge Base $\mathcal{K}$ and History Cache $\mathcal{H}$:**
  We clarified the physics-informed loss formulation across architectures, the concrete template-based code generation process, and the data-driven construction of Knowledge Base $\mathcal{K}$ together with history-based reuse through History Cache $\mathcal{H}$.

- **Symbolic–Semantic Verification Validity:**
  We added quantitative evidence demonstrating strong correlation between semantic score and downstream PINN solvability.

- **Efficiency, Scalability & Error Attribution:**
  We added wall-clock time results, clarified decision pathways in feedback loops, and showed stability across dataset scales and ambiguous language settings.


### **2.3 Clarification on the Non-Technical Rejecting Review**
One review (jqhd) was based solely on a misunderstanding regarding anonymity compliance and did not assess the paper’s technical content. The submission fully adheres to ICLR anonymity rules.

---

## **3. Final Remarks**
We have incorporated all changes into the revised manuscript. With all concerns addressed and additional experiments included, we respectfully ask for favorable consideration of our work as the first end-to-end framework that turns natural-language scientific descriptions into executable PINNs.

Thank you for your time and consideration.

Best regards and thanks,

Authors of Submission 3140

---

### Meta-Review · Area_Chair_aCHp · 2026-01-06

**Summary:**

The submission proposes Lang-PINN, an LLM-driven multi-agent workflow that turns natural-language problem descriptions into executable PINNs, and reviewers acknowledge strong empirical gains as well as the practicality of modular code generation with verification for debugging and correctness. However, weighing the discussion and the overall assessment, the paper does not clear the bar for acceptance this cycle, so I recommend rejection.

**Reviewer Concerns:**

The main concerns focus on (i) limited methodological novelty, i.e., largely orchestrating existing agentic/LLM techniques rather than introducing new core methods for PDE formulation or architecture search, and (ii) evaluation limitations, including the small PDE coverage in Task2PDE and language settings that may test narrative understanding more than scientific reasoning; reviewers also requested missing analyses such as LLM inference cost and resource usage.

**Reviewer Scores:**

Scores are overall below the acceptance threshold: one reviewer provided a strong reject (0) primarily tied to a perceived double-blind/anonymity issue, which the authors have attempted to clarify; even setting this score aside, the remaining reviewers’ ratings stay at 4 (marginally below threshold) and they do not offer clearly positive follow-up after the rebuttal/discussion that would justify upgrading their evaluations. Taken together, the overall signal remains borderline but leaning reject rather than supportive of acceptance.

---

### Decision · Program_Chairs · 2026-01-26

Reject